# Unfolded Laplacian Spectral Embedding: A Theoretically Grounded Approach to Dynamic Network Representation

Haruka Ezoe [1]   Hiroki Matsumoto [1 2]   Ryohei Hisano [1 3]

## Abstract

Dynamic relational data arise in many machine learning applications, yet their evolving structure poses challenges for learning representations that remain consistent and interpretable over time. A common approach is to learn time varying node embeddings, whose usefulness depends on well defined stability properties across nodes and across time. We introduce Unfolded Laplacian Spectral Embedding (ULSE), a principled extension of unfolded adjacency spectral embedding to normalized Laplacian operators, a setting where stability guarantees have remained out of reach. We prove that ULSE satisfies both cross-sectional and longitudinal stability under a dynamic stochastic block model. Moreover, the Laplacian formulation yields a dynamic Cheeger-type inequality linking the spectrum of the unfolded normalized Laplacian to worst case conductance over time, providing structural insight into the embeddings. Empirical results on synthetic and real-world dynamic networks validate the theory.

## 1. Introduction

Relational data arise in many machine learning applications, including recommendation systems (Raza et al., 2026), social networks (Singh et al., 2024), biological networks (Kiani et al., 2021), and knowledge graphs (Galkin et al., 2024). In many settings these relationships evolve over time, motivating methods that represent dynamic network structure for tasks such as clustering, prediction, and anomaly detection. A central approach is to learn *dynamic node embeddings*: low-dimensional representations of nodes that vary over time and summarize evolving connectivity patterns.

For temporal comparisons to be meaningful, dynamic embeddings must satisfy basic stability requirements. *Cross-sectional stability* requires that nodes with identical connectivity behavior at a fixed time be embedded identically, while *longitudinal stability* requires that a node whose connectivity behavior does not change over time be embedded consistently across snapshots. Without these properties, temporal comparisons become unstable and difficult to interpret.

Although stability is often implicitly assumed, it has only recently been formalized. Gallagher et al. (2021) introduce precise definitions of cross-sectional and longitudinal stability and show that unfolded adjacency spectral embedding (UASE) satisfies both properties by jointly factorizing adjacency matrices across time. UASE remains one of the few dynamic embedding methods with rigorous stability guarantees.

At the same time, a fundamental gap remains. Many widely used spectral operators fall outside the scope of existing stability theory. In particular, normalized Laplacians are central in spectral graph theory due to their robustness to degree heterogeneity and their connection to clustering and conductance analysis. However, naive unfolding of normalized Laplacians can induce spurious temporal drift, since degree normalization varies across time even when a node's underlying connectivity behavior remains unchanged. Existing approaches based on perturbation, dilation (Zhao et al., 2021), or supra-Laplacian constructions (Silva et al., 2018; Froyland et al., 2024) do not resolve this issue and fail to enforce node level longitudinal stability.

In this paper, we address this gap by introducing *Unfolded Laplacian Spectral Embedding* (ULSE), a dynamic embedding framework that extends the unfolding paradigm of UASE to normalized Laplacian based operators while preserving both cross-sectional and longitudinal stability under explicit and verifiable conditions. Here, "unfolding" refers to concatenating graph operators across time, rather than algorithm unrolling or a learnable neural architecture. We study two variants, ULSE-n1 and ULSE-n2, correspond-

[1]Graduate School of Information Science and Technology, The University of Tokyo [2]CaseMatch Inc.  [3]The Canon Institute for Global Studies. Correspondence to: Ryohei Hisano <hisanor@g.ecc.u-tokyo.ac.jp>.

*Proceedings of the $43^{rd}$ International Conference on Machine Learning*, Seoul, South Korea. PMLR 306, 2026. Copyright 2026 by the author(s).

ing to distinct normalization strategies across time. Under a dynamic stochastic block model (Holland et al., 1983), we prove that ULSE embeddings converge to well defined population limits and satisfy the desired stability properties.

Beyond stability guarantees, ULSE admits a structural interpretation grounded in classical spectral graph theory. For ULSE-n1, we derive a dynamic Cheeger-type inequality (Chung, 1997; Nica, 2018) linking the spectrum of the unfolded normalized Laplacian to worst case conductance across time. Since ULSE embeddings are constructed from the corresponding singular vectors, this result connects dynamic cuts, spectral values, and dynamic node embeddings within a single unfolded Laplacian framework. To our knowledge, this is the first such result for unfolded normalized-Laplacian embeddings with cross-sectional and longitudinal stability guarantees. Experiments on synthetic and real-world dynamic networks support the theory and demonstrate strong performance, particularly under degree heterogeneity.

The main contributions of this paper are:

- We introduce Unfolded Laplacian Spectral Embedding (ULSE), extending unfolded adjacency spectral embedding to normalized Laplacian based operators in dynamic graphs.

- We establish convergence and prove both cross-sectional and longitudinal stability for ULSE under a dynamic stochastic block model.

- We derive a dynamic Cheeger-type inequality connecting dynamic cuts, spectral values, and dynamic node embeddings within the unfolded normalized Laplacian framework.

- We empirically validate the stability guarantees of ULSE and demonstrate strong clustering performance across synthetic and real-world dynamic networks, particularly under degree heterogeneity.

## 2. Problem Setup and Stability Requirements

### 2.1. Dynamic Network Embedding Setup

We observe a dynamic network as a sequence of $T$ graph snapshots $\mathcal{G} = \{G^{(1)}, \ldots, G^{(T)}\}$, where each snapshot $G^{(t)}$ is defined on a common vertex set of size $n$. Let $\mathbf{A}^{(t)} \in \{0, 1\}^{n \times n}$ denote the symmetric adjacency matrix at time $t$, where $\mathbf{A}_{ij}^{(t)} = 1$ indicates the presence of an edge between nodes $i$ and $j$. Throughout, we consider undirected graphs and enforce symmetry by setting $\mathbf{A}_{ij}^{(t)} = \mathbf{A}_{ji}^{(t)}$.

Each snapshot is modeled as an inhomogeneous random graph (Jones & Rubin-Delanchy, 2020). For $i \leq j$, edges

are generated independently according to

$$\mathbf{A}_{ij}^{(t)} \sim \text{Bernoulli}\left(\mathbf{P}_{ij}^{(t)}\right),$$

where $\mathbf{P}^{(t)} \in [0, 1]^{n \times n}$ is a symmetric matrix of edge probabilities at time $t$. Conditional on $\{\mathbf{P}^{(t)}\}_{t=1}^{T}$, edges are independent across node pairs and time steps.

Given $\mathcal{G}$, our objective is to learn two types of node representations: (i) a time invariant anchor embedding $\hat{\mathbf{X}} \in \mathbb{R}^{n \times d}$, and (ii) a sequence of time varying dynamic embeddings $\hat{\mathbf{Y}}^{(1)}, \ldots, \hat{\mathbf{Y}}^{(T)} \in \mathbb{R}^{n \times d}$. The anchor embedding captures persistent node specific structure shared across time, while each dynamic embedding captures snapshot specific variation. We require that the dynamic embeddings are intrinsically aligned across time so that comparisons across snapshots are meaningful without post hoc Procrustes alignment. This alignment is achieved by constructing embeddings jointly across all snapshots.

For theoretical analysis, we also consider population (noise-free) quantities obtained by replacing $\mathbf{A}^{(t)}$ with $\mathbf{P}^{(t)}$ in the same construction. Population objects are denoted with a tilde, for example $\tilde{\mathbf{X}}$ and $\tilde{\mathbf{Y}}^{(t)}$, and similarly for degrees, Laplacian type operators, and associated spectral subspaces.

We use standard asymptotic notation $\mathcal{O}(\cdot), \Omega(\cdot), \Theta(\cdot), o(\cdot)$, and $\omega(\cdot)$, with asymptotics taken as $n \to \infty$ unless otherwise specified, and write $\text{poly}(k)$ for a polynomial in $k$. Unless otherwise stated, $\|\cdot\|_2$ denotes the spectral norm for matrices, $\|\cdot\|_F$ the Frobenius norm, and $\|\mathbf{x}\|$ the Euclidean norm of a vector $\mathbf{x}$. We repeatedly use standard norm inequalities, including $\|\mathbf{X}\|_2 \leq \|\mathbf{X}\|_F \leq \sqrt{\text{rank}(\mathbf{X})}\|\mathbf{X}\|_2$ for a matrix $\mathbf{X}$. Bounds are understood to hold almost surely. Table 2 in the appendix summarizes the notation used throughout the paper.

### 2.2. Stability Conditions

A central requirement for dynamic embeddings is that they support consistent comparisons both within a snapshot and across snapshots. Following the stability framework of Gallagher et al. (2021) (with a slightly relaxed formulation that does not require identical error distributions), we seek dynamic embeddings $\hat{\mathbf{Y}}^{(t)}$ that converge to population limits $\mathbf{Y}^{(t)}$ as $n \to \infty$, and whose population counterparts satisfy the following properties:

1. **Cross-sectional stability:** If $\mathbf{P}_{i:}^{(t)} = \mathbf{P}_{j:}^{(t)}$, then $\mathbf{Y}_{i:}^{(t)} = \mathbf{Y}_{j:}^{(t)}$.

2. **Longitudinal stability:** If $\mathbf{P}_{i:}^{(t)} = \mathbf{P}_{i:}^{(s)}$, then $\mathbf{Y}_{i:}^{(t)} = \mathbf{Y}_{i:}^{(s)}$.

Here $\mathbf{P}_{i:}^{(t)}$ denotes the $i$-th row of $\mathbf{P}^{(t)}$. Cross-sectional stability ensures that nodes with identical connectivity behavior

at a fixed time are embedded identically, while longitudinal stability ensures that a node whose connectivity behavior does not change over time receives a consistent embedding. These conditions are imposed at the population level and describe the target behavior that consistent estimators should recover asymptotically.

The stability conditions above are defined directly in terms of the underlying connectivity profiles $\mathbf{P}_{i:}^{(t)}$ and are therefore agnostic to the specific embedding construction. While unfolded adjacency based methods can be shown to satisfy both properties under mild conditions, enforcing the same guarantees for Laplacian based operators is substantially more delicate. Normalized Laplacians depend explicitly on degree matrices, so equality of probability rows does not generally imply equality of normalized rows unless the corresponding degrees are also matched. This observation highlights that longitudinal stability cannot be achieved without imposing explicit conditions on the degree structure across time. We formally establish and analyze the necessity of such conditions in later sections.

### 2.3. Dynamic Stochastic Block Model (DSBM)

To obtain explicit stability guarantees, we specialize the inhomogeneous random graph model to a dynamic stochastic block model (DSBM) (Holland et al., 1983). At each time $t$, every node $i \in \{1, \ldots, n\}$ is assigned a (possibly time varying) community label $\mathbf{z}_i^{(t)} \in \{1, \ldots, K^{(t)}\}$, where $K^{(t)}$ denotes the number of communities present at time $t$, with marginal community proportions $\boldsymbol{\pi}^{(t)} = (\pi_1^{(t)}, \ldots, \pi_{K^{(t)}}^{(t)})$ satisfying $\pi_k^{(t)} > 0$ for all $k$. Let $\mathbf{n}_k^{(t)}$ denote the number of nodes assigned to community $k$ at time $t$. We assume that the collection of label trajectories $\{\mathbf{z}_i^{(t)}\}_{t=1}^T$ admits only $K$ distinct temporal patterns. These patterns correspond to $K$ latent trajectories, each describing how a node's community membership evolves over time. Let $\boldsymbol{\pi} = (\pi_1, \ldots, \pi_K)$ denote the distribution over these latent trajectories, where each node independently follows the $k$-th trajectory with probability $\pi_k$. Accordingly, we associate each node $i \in \{1, \ldots, n\}$ with a community trajectory label $\mathbf{z}_i \in \{1, \ldots, K\}$, drawn independently according to $\boldsymbol{\pi}$. To map latent trajectories to time-specific communities, for each time $t$ we introduce a mapping

$$c^{(t)} : [K] \to [K^{(t)}],$$

where $c^{(t)}(k)$ specifies the community index at time $t$ corresponding to the $k$-th latent trajectory. Under this formulation, the community label of node $i$ at time $t$ is given by

$$\mathbf{z}_i^{(t)} = c^{(t)}(\mathbf{z}_i), \quad t = 1, \ldots, T.$$

Conditional on the community assignments at time $t$, edges are generated independently according to a sparsity param-

eter $\rho = \rho(n) \in (0, 1]$ and a symmetric block probability matrix $\mathbf{B}^{(t)} \in [0, 1]^{K^{(t)} \times K^{(t)}}$, so that

$$\mathbf{P}_{ij}^{(t)} = \rho \, \mathbf{B}_{\mathbf{z}_i^{(t)} \mathbf{z}_j^{(t)}}^{(t)}.$$

We allow either the dense regime $\rho = 1$ or a sparse regime in which $\rho \to 0$ as $n \to \infty$, and assume throughout that $\rho = \omega(\log n/n)$. In addition, we assume a strictly positive minimum block probability at each time step,

$$\mathbf{B}_{\min}^{(t)} := \min_{k,\ell \in \{1, \ldots, K^{(t)}\}} \mathbf{B}_{k\ell}^{(t)} > 0.$$

Under the DSBM, equality of connectivity profiles $\mathbf{P}_{i:}^{(t)}$ corresponds to equality of block-level connectivity patterns at time $t$, independent of the specific labels assigned to nodes at other times. This observation allows the stability conditions of Section 2.2 to be formulated and verified entirely in terms of $\mathbf{P}^{(t)}$ and $\mathbf{B}^{(t)}$, without requiring any persistence or temporal coupling of the community labels $\{\mathbf{z}_i^{(t)}\}$. Consequently, the population embeddings and finite-sample convergence guarantees for unfolded spectral methods derived in subsequent sections depend only on the sequence of edge probability matrices $\{\mathbf{P}^{(t)}\}_{t=1}^T$, and not on the temporal evolution of individual node memberships. For the theoretical analysis, equivalence classes are defined by the $K$ distinct label trajectories and have fixed sizes $\mathbf{n}$. Nodes assigned to different trajectories may coincide in the same community at a given time $t$, i.e., $\mathbf{z}_i^{(t)} = \mathbf{z}_j^{(t)}$, even though their trajectories differ over the full time horizon.

### 2.4. Unfolded Adjacency Spectral Embedding (UASE)

We briefly review unfolded adjacency spectral embedding (UASE) (Gallagher et al., 2021), which provides a baseline unfolded spectral method with established stability guarantees. The unfolded adjacency matrix is formed by concatenating the $T$ adjacency matrices horizontally:

$$\mathcal{A} = [\mathbf{A}^{(1)} \mid \cdots \mid \mathbf{A}^{(T)}] \in \mathbb{R}^{n \times nT}.$$

Let $\mathcal{A}$ admit a rank-$d$ truncated singular value decomposition $\mathcal{A} = \mathbf{U}\boldsymbol{\Sigma}\mathbf{V}^\top + \mathbf{U}_\perp\boldsymbol{\Sigma}_\perp\mathbf{V}_\perp^\top$, where $\mathbf{U} \in \mathbb{O}(n \times d)$, $\mathbf{V} \in \mathbb{O}(nT \times d)$, and $\boldsymbol{\Sigma} \in \mathbb{R}^{d \times d}$ contains the $d$ largest singular values. Here, $\mathbb{O}(n \times d)$ denotes the set of all real $n \times d$ matrices with orthonormal columns, that is, $\mathbb{O}(n \times d) = \{\mathbf{U} \in \mathbb{R}^{n \times d} : \mathbf{U}^\top\mathbf{U} = \mathbf{I}_d\}$. Partition $\mathbf{V}$ into $T$ consecutive blocks $\mathbf{V}^{(t)} \in \mathbb{R}^{n \times d}$. UASE defines the anchor and dynamic embeddings as

$$\hat{\mathbf{X}} = \mathbf{U}\boldsymbol{\Sigma}^{1/2}, \qquad \hat{\mathbf{Y}}^{(t)} = \mathbf{V}^{(t)}\boldsymbol{\Sigma}^{1/2}, \quad t = 1, \ldots, T.$$

UASE is one of the few dynamic embedding constructions for which both cross-sectional and longitudinal stability can

be established under an explicit probabilistic model. By jointly factorizing all snapshots, the unfolding construction intrinsically aligns embeddings across time and avoids the need for post hoc alignment procedures. UASE therefore serves as the conceptual and technical foundation for our goal of extending the unfolding paradigm to Laplacian based operators while preserving the same stability requirements.

# 3. Unfolded Laplacian Spectral Embedding

This section introduces *Unfolded Laplacian Spectral Embedding* (ULSE), a dynamic network embedding framework that extends unfolded adjacency spectral embedding (UASE) to normalized Laplacian type operators. ULSE produces both a time invariant anchor embedding and a sequence of time dependent dynamic embeddings, while preserving the stability guarantees established theoretically in Section 4.

## 3.1. Degree Quantities and Notation

For each snapshot $t \in \{1, \ldots, T\}$, define the degree vector and degree matrix

$$\mathbf{d}^{(t)} = \mathbf{A}^{(t)}\mathbf{1}, \qquad \mathbf{D}^{(t)} = \operatorname{diag}\left(\mathbf{d}^{(t)}\right),$$

where $\mathbf{1} \in \mathbb{R}^n$ denotes the all-ones vector and $\mathbf{d}_i^{(t)}$ is the degree of node $i$ at time $t$. We also define the aggregated degree quantities across time,

$$\mathbf{d}^{(1:T)} = \sum_{t=1}^{T} \mathbf{d}^{(t)}, \qquad \mathbf{D}^{(1:T)} = \operatorname{diag}\left(\mathbf{d}^{(1:T)}\right).$$

Throughout, we assume that the degree matrices used for normalization are invertible. Under the DSBM and the assumed sparsity regime this holds with high probability, and in practice diagonal regularization may be applied when necessary.

## 3.2. Unfolding Normalized Laplacian Operators

ULSE replaces the adjacency matrices used in UASE with normalized Laplacian type operators and constructs a joint embedding across all time steps by horizontally unfolding these operators. We consider two normalization strategies, leading to two variants: *ULSE-n1* and *ULSE-n2*. The two variants differ only in how degree normalization is handled across time and are designed to address distinct sources of instability. For either variant, we form an unfolded matrix of the form

$$\mathcal{L} = \left[\mathbf{L}^{(1)} \mid \cdots \mid \mathbf{L}^{(T)}\right] \in \mathbb{R}^{n \times nT},$$

and compute a rank-$d$ truncated singular value decomposition, yielding an anchor embedding shared across time and a sequence of snapshot specific dynamic embeddings that are intrinsically aligned.

## 3.3. ULSE-n1: Per-Snapshot Normalization

ULSE-n1 applies normalization independently at each time step. For each snapshot $t$, define the normalized Laplacian

$$\mathbf{L}_{\mathrm{n1}}^{(t)} = \mathbf{I} - \mathbf{D}^{(t)-1/2}\mathbf{A}^{(t)}\mathbf{D}^{(t)-1/2}.$$

The unfolded normalized Laplacian is then

$$\mathcal{L}_{\mathrm{n1}} = \left[\mathbf{L}_{\mathrm{n1}}^{(1)} \mid \cdots \mid \mathbf{L}_{\mathrm{n1}}^{(T)}\right].$$

We compute a singular value decomposition

$$\mathcal{L}_{\mathrm{n1}} = \mathbf{U}\boldsymbol{\Sigma}\mathbf{V}^{\top} + \mathbf{U}_{\perp}\boldsymbol{\Sigma}_{\perp}\mathbf{V}_{\perp}^{\top},$$

where $\mathbf{U} \in \mathbb{O}^{n \times d}$, $\mathbf{V} \in \mathbb{O}^{nT \times d}$, and let $\boldsymbol{\Sigma} \in \mathbb{R}^{d \times d}$ be the diagonal matrix containing the singular values $\sigma_2, \ldots, \sigma_{d+1}$ of $\mathcal{L}_{\mathrm{n1}}$, ordered increasingly. Although these singular values may in principle be zero, we assume they are positive. Partition $\mathbf{V}$ into $T$ consecutive blocks $\mathbf{V}^{(t)} \in \mathbb{R}^{n \times d}$. The anchor embedding and dynamic embeddings are defined as

$$\hat{\mathbf{X}} = \mathbf{U}\boldsymbol{\Sigma}^{1/2},$$
$$\hat{\mathbf{Y}}^{(t)} = \mathbf{V}^{(t)}\boldsymbol{\Sigma}^{1/2} - \mathbf{U}\boldsymbol{\Sigma}^{-1/2}, \quad t = 1, \ldots, T.$$

The subtraction of $\mathbf{U}\boldsymbol{\Sigma}^{-1/2}$ is essential for achieving cross-sectional stability under normalized Laplacian embeddings and admits a direct operator level interpretation at the population level. Since each block satisfies $\mathbf{L}_{\mathrm{n1}}^{(t)} = \mathbf{I} - \mathbf{D}^{(t)-1/2}\mathbf{A}^{(t)}\mathbf{D}^{(t)-1/2}$, the reconstructed embedding $\hat{\mathbf{Y}}^{(t)}$ contains a node specific shift induced by aggregating information across all snapshots, arising from the identity component. ULSE-n1 explicitly removes this node wise shift, thereby restoring cross-sectional stability so that each snapshot embedding reflects only contemporaneous degree-normalized connectivity.

## 3.4. ULSE-n2: Partially Aggregated Normalization

ULSE-n2 incorporates time aggregated degree information into the normalization in order to enforce longitudinal stability without requiring an explicit correction term. For each snapshot $t$, we define the partially normalized Laplacian type operator

$$\mathbf{L}_{\mathrm{n2}}^{(t)} = -\mathbf{D}^{(1:T)-1/2}\mathbf{A}^{(t)}\mathbf{D}^{(t)-1/2},$$

where $\mathbf{D}^{(1:T)}$ denotes the degree matrix aggregated across all time steps. Unlike the standard normalized Laplacian, $\mathbf{L}_{\mathrm{n2}}^{(t)}$ couples snapshot specific adjacency information with a global, time aggregated degree normalization on the left and a snapshot specific normalization on the right, thereby fixing node wise scaling across time.

The unfolded operator is constructed by horizontal concatenation,

$$\mathcal{L}_{\mathrm{n2}} = \left[\mathbf{L}_{\mathrm{n2}}^{(1)} \mid \cdots \mid \mathbf{L}_{\mathrm{n2}}^{(T)}\right] \in \mathbb{R}^{n \times nT}.$$

We compute a truncated singular value decomposition

$$\mathcal{L}_{\mathrm{n2}} = \mathbf{U}\boldsymbol{\Sigma}\mathbf{V}^\top + \mathbf{U}_\perp \boldsymbol{\Sigma}_\perp \mathbf{V}_\perp^\top,$$

where $\boldsymbol{\Sigma}$ contains the $d$ *largest* singular values of $\mathcal{L}_{\mathrm{n2}}$. Partitioning $\mathbf{V}$ into $T$ consecutive blocks $\mathbf{V}^{(t)} \in \mathbb{R}^{n \times d}$, the anchor embedding and dynamic embeddings are defined as

$$\hat{\mathbf{X}} = \mathbf{U}\boldsymbol{\Sigma}^{1/2}, \qquad \hat{\mathbf{Y}}^{(t)} = \mathbf{V}^{(t)}\boldsymbol{\Sigma}^{1/2}, \quad t = 1, \dots, T.$$

In contrast to ULSE-n1, ULSE-n2 does not require an explicit centering or correction term. The use of a common left normalization by $\mathbf{D}^{(1:T)}$ ensures that the scale of each node's embedding is fixed across time, while the snapshot specific right normalization preserves sensitivity to temporal variation. As a result, nodes whose connectivity profiles remain unchanged across snapshots are embedded consistently, guaranteeing longitudinal stability under the dynamic stochastic block model without requiring equality of snapshot specific degrees.

ULSE is a nonparametric spectral method whose dominant cost is a truncated SVD of the unfolded matrix. With $T$ snapshots on $n$ nodes, the unfolded matrix has size $n \times nT$, and a $d$-dimensional embedding is obtained from its rank-$d$ truncated SVD. Thus, ULSE has essentially the same computational structure as UASE: there is no iterative training or learned-parameter optimization, and the computational bottleneck is a single spectral decomposition. In practice, standard sparse or iterative truncated-SVD routines can be used, so the runtime is governed by repeated matrix-vector products and depends primarily on the target dimension $d$ and the sparsity of the unfolded matrix. The two variants, ULSE-n1 and ULSE-n2, differ only in how the unfolded matrix is normalized, not in the overall computational bottleneck.

The two ULSE variants are intended for complementary regimes. ULSE-n1 is closest to the classical normalized Laplacian and therefore retains a direct connection to standard spectral clustering and Cheeger-type theory. Its longitudinal stability, however, requires the relevant population degree structure to remain unchanged across time, reflecting the fact that per-snapshot normalization can introduce temporal rescaling. ULSE-n2 is designed to remove this obstruction by using a common time-aggregated left normalization, and is therefore preferable when degree levels vary substantially across snapshots but node connectivity profiles should still be compared over time. In practice, ULSE-n1 is useful when preserving the standard normalized-Laplacian interpretation is important, whereas ULSE-n2 is the more robust choice when temporal degree variability or a heterophilous regime is expected.

## 4. Theoretical Guarantees

This section presents the main theoretical guarantees for Unfolded Laplacian Spectral Embedding (ULSE). Under the dynamic stochastic block model introduced in Section 2.3, we show that ULSE produces dynamic embeddings that (i) converge to well defined population (noise free) limits and (ii) satisfy both cross-sectional and longitudinal stability as defined in Section 2.2. Results are established separately for ULSE-n1 and ULSE-n2, followed by a structural interpretation based on a dynamic Cheeger-type inequality.

### 4.1. Stability Guarantees for ULSE-n1

We work under the dynamic stochastic block model introduced in Section 2.3 and the notation established in Sections 2 and 3. Let $\mathbf{P}^{(t)}$ denote the edge probability matrix at time $t$, and let $\tilde{\mathbf{L}}_{\mathrm{n1}}^{(t)}$ be the corresponding population normalized Laplacian defined using $\mathbf{P}^{(t)}$ and the population degree matrix $\tilde{\mathbf{D}}^{(t)}$. Define the unfolded population operator

$$\tilde{\mathcal{L}}_{\mathrm{n1}} = \big[\tilde{\mathbf{L}}_{\mathrm{n1}}^{(1)} \mid \cdots \mid \tilde{\mathbf{L}}_{\mathrm{n1}}^{(T)}\big].$$

To characterize the informative spectral structure of $\tilde{\mathcal{L}}_{\mathrm{n1}}$, we also consider the associated community level operators. Let $\tilde{\mathbf{Q}}^{(t)} \in [0,1]^{K \times K}$ and $\bar{\mathbf{Q}}^{(t)} \in [0,1]^{K \times K}$ be the community level edge probability matrices at time $t$, defined as

$$\tilde{\mathbf{Q}}_{kl}^{(t)} = \frac{n_k}{n}\frac{n_l}{n}\mathbf{B}_{c^{(t)}(k)c^{(t)}(l)}^{(t)}, \qquad \bar{\mathbf{Q}}_{kl}^{(t)} = \pi_k \pi_l \mathbf{B}_{c^{(t)}(k)c^{(t)}(l)}^{(t)}.$$

Let $\tilde{\mathbf{M}}_{\mathrm{n1}}^{(t)}$ and $\bar{\mathbf{M}}_{\mathrm{n1}}^{(t)}$ be the normalized Laplacians constructed from $\tilde{\mathbf{Q}}^{(t)}$ and $\bar{\mathbf{Q}}^{(t)}$, respectively. We further define $\tilde{\mathcal{M}}_{\mathrm{n1}}$ and $\bar{\mathcal{M}}_{\mathrm{n1}}$ as the unfolded normalized Laplacians, and denote their singular values as follows

$$\tilde{\lambda}_1 \le \tilde{\lambda}_2 \le \cdots \le \tilde{\lambda}_K, \quad \bar{\lambda}_1 \le \bar{\lambda}_2 \le \cdots \le \bar{\lambda}_K.$$

The results below show that ULSE-n1 recovers stable dynamic embeddings provided that these informative singular values are well separated from the high multiplicity singular value $\sqrt{T}$ induced by the identity components of the normalized Laplacian. This separation condition ensures that the informative bottom singular subspace of $\tilde{\mathcal{L}}_{\mathrm{n1}}$ is identifiable and robust to sampling noise.

**Theorem 4.1** (Stability of ULSE-n1). *Suppose that the community level singular values satisfy*

$$0 \le \bar{\lambda}_1 < \bar{\lambda}_2 \le \cdots \le \bar{\lambda}_K < \sqrt{T}.$$

*Set the embedding dimension $d = K - 1$. Then there exist matrices $\mathbf{Y}^{(t)} \in \mathbb{R}^{n \times d}$, $t = 1, \dots, T$, such that*

$$\max_{t \in \{1, \dots, T\}} \big\|\hat{\mathbf{Y}}^{(t)} - \mathbf{Y}^{(t)}\big\|_2 = \mathcal{O}\Big((\rho n)^{-1/2}\Big) \quad a.s.,$$

*and the following stability properties hold:*

- **Cross-sectional stability:** If $\mathbf{P}_{i:}^{(t)} = \mathbf{P}_{j:}^{(t)}$, then $\mathbf{Y}_{i:}^{(t)} = \mathbf{Y}_{j:}^{(t)}$.

- **Longitudinal stability:** If $\mathbf{P}_{i:}^{(t)} = \mathbf{P}_{i:}^{(s)}$ and $\tilde{\mathbf{D}}^{(t)} = \tilde{\mathbf{D}}^{(s)}$, then $\mathbf{Y}_{i:}^{(t)} = \mathbf{Y}_{i:}^{(s)}$.

Theorem 4.1 establishes that ULSE-n1 produces stable dynamic embeddings. Cross-sectional stability follows from invariance of population connectivity profiles, while longitudinal stability is guaranteed when population degree matrices remain unchanged across time. This requirement is not an artifact of the analysis but a fundamental consequence of per-snapshot left degree normalization: no method based on independently left normalized Laplacians can achieve exact longitudinal invariance when degrees vary over time. ULSE-n1 therefore characterizes the strongest form of stability achievable under per-snapshot left side normalization and serves as a theoretically transparent baseline.

The proof follows a two step argument. We first establish uniform convergence of the empirical ULSE-n1 embeddings to their population counterparts at rate $(\rho n)^{-1/2}$. We then show that the population embeddings satisfy the desired stability properties exactly. Combining these two results yields the stated stability guarantees.

**Theorem 4.2** (Convergence of ULSE-n1). *Assume $d = K - 1$. There exists an orthogonal matrix $\mathbf{W} \in \mathbb{O}(d \times d)$ such that, for each $t \in \{1, \ldots, T\}$,*

$$\|\hat{\mathbf{Y}}^{(t)} - \tilde{\mathbf{Y}}^{(t)}\mathbf{W}\|_2 = \mathcal{O}\left((\rho n)^{-1/2}\right) \quad a.s.$$

The bound in Theorem 4.2 implies consistency of the empirical embeddings. Here, "uniform" refers to control that holds over all time steps $t \in \{1, \ldots, T\}$, while the error within each snapshot is aggregated across nodes through the spectral norm. This differs from stronger row-wise notions of consistency, such as two-to-infinity consistency. When combined with the population level stability properties of $\tilde{\mathbf{Y}}^{(t)}$ established below, this yields the stability guarantee stated in Theorem 4.3.

**Theorem 4.3** (Stability of noise-free ULSE-n1). *The population embeddings $\tilde{\mathbf{Y}}^{(t)}$ satisfy:*

- *If $\mathbf{P}_{i:}^{(t)} = \mathbf{P}_{j:}^{(t)}$, then $\tilde{\mathbf{Y}}_{i:}^{(t)} = \tilde{\mathbf{Y}}_{j:}^{(t)}$.*

- *If $\mathbf{P}_{i:}^{(t)} = \mathbf{P}_{i:}^{(s)}$ and $\tilde{\mathbf{D}}^{(t)} = \tilde{\mathbf{D}}^{(s)}$, then $\tilde{\mathbf{Y}}_{i:}^{(t)} = \tilde{\mathbf{Y}}_{i:}^{(s)}$.*

Setting $\mathbf{Y}^{(t)} = \tilde{\mathbf{Y}}^{(t)}\mathbf{W}$ and applying a union bound over $t \in \{1, \ldots, T\}$ completes the proof of Theorem 4.1. To establish Theorem 4.2, we analyze the unfolded normalized Laplacian operator using a standard spectral perturbation strategy. The proof proceeds in three steps: we first bound the deviation between the empirical unfolded operator $\mathcal{L}_{\mathrm{n1}}$

and its population counterpart $\tilde{\mathcal{L}}_{\mathrm{n1}}$ (Lemma 4.4); we then characterize the singular value structure of $\tilde{\mathcal{L}}_{\mathrm{n1}}$ and identify the informative subspace (Lemma 4.5 and Corollary 4.6); finally, we apply a Davis–Kahan perturbation argument combined with row-wise norm control to bound the deviation of the empirical singular subspaces (Lemma 4.7).

**Lemma 4.4** (Deviation bound).

$$\|\mathcal{L}_{\mathrm{n1}} - \tilde{\mathcal{L}}_{\mathrm{n1}}\|_2 = \mathcal{O}\left((\rho n)^{-1/2}\right) \quad a.s.$$

This bound shows that unfolding per-snapshot normalized Laplacians does not amplify sampling noise: the empirical unfolded operator remains a small perturbation of its noise-free counterpart, even after concatenation across time.

Let $\mathcal{Z} \in \mathbb{R}^{n \times K}$ denote the indicator matrix of equivalence classes of nodes with identical population connectivity profiles across all time steps, and let $\mathcal{S} = \mathrm{span}(\mathcal{Z})$.

**Lemma 4.5** (Spectral structure of the population operator). *Assume the community size matrix $\mathbf{N} = \mathrm{diag}(\mathbf{n})$ is invertible. Then $\tilde{\mathcal{L}}_{\mathrm{n1}}$ has (i) a singular value $\sqrt{T}$ with multiplicity $n - K$, whose left singular vectors span $\mathcal{S}^{\perp}$, and (ii) $K$ remaining singular values $\tilde{\lambda}_1, \ldots, \tilde{\lambda}_K$, whose left singular vectors lie in $\mathcal{S}$.*

Lemma 4.5 implies that all informative population level structure is confined to a $K$-dimensional subspace, while the orthogonal complement corresponds to a high multiplicity, non-informative singular value induced by the identity components of the normalized Laplacian.

**Corollary 4.6.** *Under the separation condition on $\bar{\lambda}_1, \ldots, \bar{\lambda}_K$, the singular values of $\mathcal{L}_{\mathrm{n1}}$ satisfy*

$$\sigma_n = \mathcal{O}(1), \qquad \sigma_2 - \sigma_1 = \Omega(1),$$
$$\sigma_{K+1} - \sigma_K = \Omega(1) \quad a.s.$$

The constant spectral gaps in this corollary ensure that the informative singular subspace is well separated from the remainder of the spectrum, which is essential for the stability of the subsequent subspace perturbation analysis.

**Lemma 4.7** (Projection deviation bound).

$$\|\mathbf{U}\mathbf{U}^{\top} - \tilde{\mathbf{U}}\tilde{\mathbf{U}}^{\top}\|_F = \mathcal{O}\left((\rho n)^{-1/2}\right) \quad a.s.,$$
$$\|\mathbf{V}\mathbf{V}^{\top} - \tilde{\mathbf{V}}\tilde{\mathbf{V}}^{\top}\|_F = \mathcal{O}\left((\rho n)^{-1/2}\right) \quad a.s.$$

Lemma 4.7 converts the operator level deviation and spectral separation into quantitative control of the empirical singular subspaces, enabling uniform convergence of the resulting node embeddings.

### 4.2. Stability Guarantees for ULSE-n2

ULSE-n2 employs partially time aggregated degree normalization, which removes the need for per-snapshot degree

matching in order to achieve longitudinal stability. Let $\mathcal{L}_{n2}$ denote the unfolded operator constructed using the ULSE-n2 normalization, and let $\bar{\mathcal{L}}_{n2}$ denote its community level population counterpart under the dynamic stochastic block model.

**Theorem 4.8** (Stability of ULSE-n2). *Suppose that the community level operator $\bar{\mathcal{L}}_{n2}$ has rank $K$, and set the embedding dimension $d = K$. Then there exist matrices $\mathbf{Y}^{(t)} \in \mathbb{R}^{n \times d}$, $t = 1, \ldots, T$, such that $\max_{t \in \{1,\ldots,T\}} \|\hat{\mathbf{Y}}^{(t)} - \mathbf{Y}^{(t)}\|_2 = \mathcal{O}((\rho n)^{-1/2})$ almost surely, and the following stability properties hold:*

- ***Cross-sectional stability:*** *If $\mathbf{P}_{i:}^{(t)} = \mathbf{P}_{j:}^{(t)}$, then $\mathbf{Y}_{i:}^{(t)} = \mathbf{Y}_{j:}^{(t)}$.*

- ***Longitudinal stability:*** *If $\mathbf{P}_{i:}^{(t)} = \mathbf{P}_{i:}^{(s)}$, then $\mathbf{Y}_{i:}^{(t)} = \mathbf{Y}_{i:}^{(s)}$.*

In contrast to ULSE-n1, ULSE-n2 retains all $K$ informative dimensions and does not require equality of snapshot specific degree matrices to ensure longitudinal stability. By using a common, time aggregated degree normalization, ULSE-n2 fixes node wise scaling across snapshots at the population level, so that nodes with unchanged connectivity profiles are embedded consistently over time. Since the proof is analogous to that of ULSE-n1, we omit the details.

### 4.3. Structural Interpretation via a Dynamic Cheeger Inequality

Beyond stability guarantees, ULSE admits a structural interpretation that links its spectrum to graph partition quality over time. Let $\mathcal{G} = \{G^{(1)}, \ldots, G^{(T)}\}$ denote the dynamic graph, and define its dynamic $k$-way conductance by

$$\phi_k(\mathcal{G}) = \max_{t \in \{1,\ldots,T\}} \phi_k(G^{(t)}),$$

where $\phi_k(G^{(t)})$ is the standard $k$-way conductance of snapshot $G^{(t)}$.

Let $\sigma_k$ denote the $k$-th smallest nontrivial singular value of the unfolded normalized Laplacian $\mathcal{L}_{n1}$. The following result relates the spectrum of the unfolded operator to the worst case partition quality across time.

**Proposition 4.9** (Dynamic Cheeger bound).

$$\frac{\sqrt{\max\left\{\sigma_k^2 - \min_{t \in \{1,\ldots,T\}} \left\|\mathcal{L}_{n1}^{-t}\right\|_2^2, 0\right\}}}{2} \leq \phi_k(\mathcal{G}) \leq \mathrm{poly}(k)\sqrt{\sigma_k},$$

*where $\mathcal{L}_{n1}^{-t}$ denotes the unfolded normalized Laplacian with the block corresponding to snapshot $t$ removed.*

Proposition 4.9 provides a dynamic analogue of the higher order Cheeger inequality. The lower bound quantifies the extent to which the $k$-th singular structure of the unfolded operator is supported uniformly across time. If for all snapshots $t$,

$$\sigma_k^2 \leq \left\|\mathcal{L}_{n1}^{-t}\right\|_2^2,$$

then the lower bound becomes vacuous. In this case, the unfolded spectrum cannot rule out the possibility of a low conductance $k$-way partition shared across time.

The upper bound follows by applying the higher order Cheeger inequality to each snapshot individually and taking a worst case maximum over time. Together, the bounds show that small singular values of the unfolded normalized Laplacian correspond to the existence of partitions that achieve low conductance simultaneously across snapshots, while large singular values indicate temporal inconsistency in partition structure. The proof combines Weyl's inequality with the higher order Cheeger inequality and is provided in the appendix.

## 5. Related Work

Dynamic network embedding methods aim to learn low-dimensional representations of evolving graphs that are comparable across time. Early approaches embed snapshots independently and rely on post hoc alignment, which does not guarantee intrinsic temporal consistency.

Joint spectral constructions provide a more principled alternative. Unfolded adjacency spectral embedding (UASE) (Gallagher et al., 2021) is one of the few methods with formal cross-sectional and longitudinal stability guarantees. Ceccherini et al. (2025) study attributed dynamic network embeddings with stability guarantees, whereas ULSE focuses on purely structural dynamic embeddings and addresses a distinct technical gap: the absence of a normalized-Laplacian analogue of unfolded spectral embedding with provable cross-sectional and longitudinal stability. However, existing theory does not extend to normalized Laplacian operators or account for the role of degree normalization in dynamic settings.

Deep learning approaches for temporal graphs, including JODIE (Kumar et al., 2019), DyRep (Trivedi et al., 2019), TGN (Rossi et al., 2020), and DyGFormer (Yu et al., 2023), achieve strong predictive performance but lack stability guarantees for time varying embeddings.

Normalized Laplacians are central in static spectral graph theory (Chung, 1997) and have been extended to temporal graphs via supra-Laplacian constructions (Silva et al., 2018; Froyland et al., 2024), but these methods do not enforce node level temporal stability. ULSE fills this gap by providing a Laplacian based unfolded embedding with explicit stability guarantees.

# 6. Experiments

We evaluate ULSE with two goals: (i) validating the cross-sectional and longitudinal stability properties predicted by our theory, and (ii) assessing downstream utility via node clustering, with particular emphasis on robustness under degree heterogeneity. We focus on clustering because it most directly evaluates the stability properties established in our theory.

## 6.1. Datasets

We construct two synthetic dynamic networks from a dynamic stochastic block model observed over $T = 3$ time steps. Node memberships are governed by $K = 3$ latent community trajectories that map deterministically to observable communities, with the number of communities varying over time ($K^{(1)} = 3, K^{(2)} = 2, K^{(3)} = 2$). The two synthetic settings differ only in community size distributions, inducing substantially different levels of degree heterogeneity, with the second setting exhibiting a pronounced small community. We fix the intra- and inter-community probabilities at $p = 0.4$ and $q = 0.2$, respectively. Ground truth community labels are available at each time step. Additional details and extended results are provided in the appendix.

We evaluate ULSE on three real-world dynamic graph datasets: **Brain** (dynamic brain connectivity), **School** (student interaction), and **Stock** (financial dependency network). We refer to Liu et al. (2024) for detailed descriptions of the Brain and School datasets, and discretize the School data into five snapshots using equal length time bins. For the Stock dataset (Silva et al., 2018), we construct one graph per quarter from S&P 500 log returns using a nonparanormal transformation (Liu et al., 2009) and graphical lasso with a fixed regularization parameter $\lambda = 10^{-5}$ (Friedman et al., 2008).

## 6.2. Experimental Setup and Stability Diagnostics

We apply $K$-means clustering to the node embeddings at each time step (following Cheng et al. (2025)) and evaluate performance using accuracy (ACC), normalized mutual information (NMI), adjusted Rand index (ARI), and macro-averaged F1 score (F1). The source code is publicly available at https://github.com/hisanor013/ULSE, and implementation details are provided in the supplementary material.

For ULSE and other spectral baselines, the embedding dimension is selected using a standard eigengap heuristic. In the synthetic experiments, where the latent rank is known, we use the theoretical choices $d = K - 1$ for ULSE-n1 and $d = K$ for ULSE-n2. For real-world datasets, where the population rank is unknown, we select the dimension from the empirical spectrum and report additional sensitivity

analyses in the appendix.

For neural temporal-graph baselines, we use the official implementations and follow the publicly documented evaluation protocols, including the recommended or default hyperparameter settings. This protocol provides a reproducible comparison against standard baseline configurations and avoids introducing additional dataset-specific tuning degrees of freedom across methods with different objectives and different numbers of tunable parameters. Key hyperparameters are reported in Appendix J.1.

We use the synthetic dataset to empirically verify that ULSE satisfies both cross-sectional and longitudinal stability, whereas supra-Laplacian embeddings may not (Silva et al., 2018). Figure 1 visualizes embeddings across two time steps for ULSE-n1, ULSE-n2, and TempCut-N. Both ULSE variants preserve consistent representations for nodes whose connectivity profiles remain unchanged across time, in accordance with the stability requirements. In contrast, TempCut-N fails to maintain alignment across snapshots.

## 6.3. Node Clustering Results

Table 1 reports node clustering performance across all datasets, with clustering metrics averaged over all snapshots. ULSE-n1 and ULSE-n2 are consistently among the top performing methods on both synthetic and real-world graphs. In balanced settings, ULSE performs comparably to adjacency based spectral methods such as UASE, while in the presence of substantial degree heterogeneity ULSE significantly outperforms all baselines across evaluation metrics. On real-world datasets, ULSE variants achieve strong and stable performance and consistently outperform supra-Laplacian methods, which perform poorly on stability-sensitive structure.

The performance gap observed on Synthetic 2 highlights a regime in which Laplacian normalization is particularly beneficial. The presence of a substantially smaller community induces systematic degree imbalance, causing adjacency based spectral methods to suppress information from low-degree nodes and distort community recovery (Krzakala et al., 2013). In contrast, ULSE explicitly normalizes by degree, yielding substantially improved robustness under degree heterogeneity. While deep temporal models such as JODIE, DyRep, TGN, and DyGFormer can be competitive on some datasets, they do not enforce stability by design. Additional qualitative analyses in the appendix show that these methods often fail to preserve consistent community structure across time, unlike ULSE-n1 and ULSE-n2. The displacement-based diagnostics in Appendix K further show that ULSE-n2 yields stronger longitudinal stability than ULSE-n1 under temporal degree variability and heterophilous regimes.

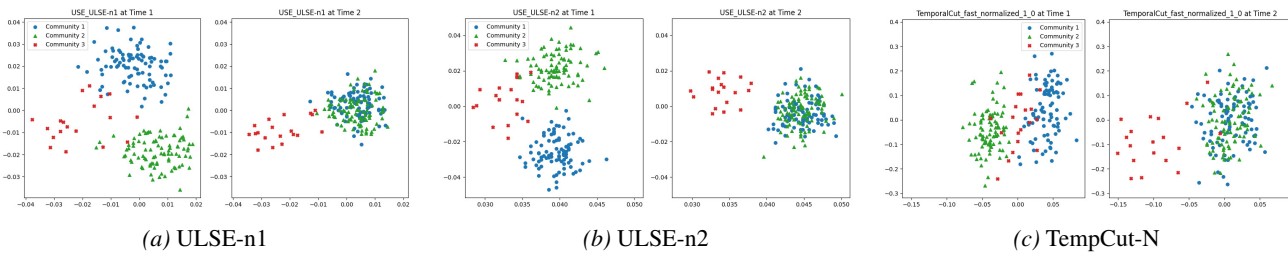

| (a) ULSE-n1 | (b) ULSE-n2 | (c) TempCut-N |

*Figure 1.* Comparison of ULSE-n1, ULSE-n2, and TempCut-N (the supra-Laplacian embedding method of Silva et al. (2018)) on the synthetic dataset using the first two embedding dimensions. The synthetic experiment consists of three snapshots, and the first two time points are visualized here for clarity. Nodes with the same color and marker belong to communities whose connectivity patterns evolve identically over time. Longitudinal stability is illustrated by Community 3 (red crosses), whose connectivity pattern remains unchanged between Times 1 and 2 and should therefore preserve its embedding position across time. Cross-sectional stability is illustrated by Communities 1 (blue circles) and 2 (green triangles), which become connectivity-equivalent at Time 2 and should therefore coincide in the embedding space. Both ULSE variants satisfy these stability properties, whereas TempCut-N fails to preserve longitudinal stability across snapshots.

| Dataset | Metric | OMNI | UASE | TempCut-N | TempCut-S | Node2Vec | JODIE | DyRep | TGN | DyGFormer | ULSE-n1 | ULSE-n2 |
|---|---|---|---|---|---|---|---|---|---|---|---|---|
| Brain | ACC | 0.246 | 0.324 | 0.129 | 0.131 | 0.303 | 0.205 | 0.179 | 0.228 | 0.157 | **0.376** | 0.358 |
| Brain | NMI | 0.196 | 0.291 | 0.003 | 0.005 | 0.220 | 0.110 | 0.065 | 0.152 | 0.028 | **0.292** | 0.283 |
| Brain | ARI | 0.087 | 0.161 | -0.000 | 0.001 | 0.103 | 0.045 | 0.031 | 0.065 | 0.012 | **0.164** | 0.153 |
| Brain | F1 | 0.201 | 0.309 | 0.111 | 0.111 | 0.334 | 0.200 | 0.159 | 0.214 | 0.140 | **0.387** | 0.362 |
| School | ACC | 0.411 | 0.853 | 0.201 | 0.204 | 0.978 | 0.194 | 0.194 | 0.216 | 0.176 | 0.994 | **0.997** |
| School | NMI | 0.472 | 0.813 | 0.066 | 0.066 | 0.965 | 0.089 | 0.085 | 0.108 | 0.056 | 0.987 | **0.994** |
| School | ARI | 0.250 | 0.655 | 0.011 | 0.010 | 0.954 | 0.028 | 0.020 | 0.024 | 0.006 | 0.985 | **0.993** |
| School | F1 | 0.360 | 0.867 | 0.188 | 0.194 | 0.977 | 0.148 | 0.154 | 0.202 | 0.109 | 0.994 | **0.997** |
| Stock | ACC | 0.225 | 0.389 | 0.170 | 0.168 | 0.247 | 0.173 | 0.213 | 0.192 | 0.156 | **0.453** | 0.410 |
| Stock | NMI | 0.125 | 0.300 | 0.069 | 0.059 | 0.185 | 0.060 | 0.109 | 0.106 | 0.000 | **0.385** | 0.337 |
| Stock | ARI | 0.038 | 0.155 | 0.006 | 0.002 | 0.065 | 0.002 | 0.028 | 0.022 | 0.000 | **0.185** | 0.154 |
| Stock | F1 | 0.177 | 0.412 | 0.156 | 0.149 | 0.281 | 0.159 | 0.180 | 0.175 | 0.036 | **0.477** | 0.430 |
| Synthetic 1 | ACC | 0.780 | 0.997 | 0.587 | 0.557 | 0.695 | 0.512 | 0.497 | 0.510 | 0.500 | **1.000** | 0.997 |
| Synthetic 1 | NMI | 0.442 | 0.984 | 0.078 | 0.051 | 0.269 | 0.007 | 0.013 | 0.023 | 0.011 | **1.000** | 0.977 |
| Synthetic 1 | ARI | 0.489 | 0.990 | 0.074 | 0.039 | 0.313 | 0.008 | 0.008 | 0.019 | 0.008 | **1.000** | 0.988 |
| Synthetic 1 | F1 | 0.774 | 0.997 | 0.578 | 0.543 | 0.688 | 0.475 | 0.460 | 0.496 | 0.480 | **1.000** | 0.997 |
| Synthetic 2 | ACC | 0.765 | 0.772 | 0.563 | 0.508 | 0.630 | 0.532 | 0.522 | 0.575 | 0.478 | 0.980 | **0.983** |
| Synthetic 2 | NMI | 0.482 | 0.551 | 0.076 | 0.008 | 0.169 | 0.003 | 0.057 | 0.104 | 0.004 | 0.865 | **0.936** |
| Synthetic 2 | ARI | 0.463 | 0.542 | 0.066 | -0.000 | 0.198 | 0.001 | 0.042 | 0.086 | -0.005 | 0.922 | **0.964** |
| Synthetic 2 | F1 | 0.786 | 0.670 | 0.521 | 0.434 | 0.560 | 0.476 | 0.488 | 0.497 | 0.419 | 0.949 | **0.966** |

*Table 1.* Node clustering results across datasets. Best results are in bold, and second best are underlined.

## 7. Conclusion

We introduced Unfolded Laplacian Spectral Embedding (ULSE), a principled framework for dynamic network representation based on normalized Laplacians. ULSE satisfies both cross-sectional and longitudinal stability under a dynamic stochastic block model, extending prior guarantees for unfolded adjacency embeddings to a broader class of spectral operators.

Our analysis yields a dynamic Cheeger-type inequality linking the spectrum of the unfolded normalized Laplacian to conductance over time, and experiments support the theory and demonstrate strong clustering performance, particularly under degree heterogeneity. Together, these results establish ULSE as a stability guaranteed unfolded Laplacian framework that connects dynamic cuts, spectral values, and dynamic node embeddings while retaining the benefits of normalized Laplacian representations.

Future work includes extensions to directed and weighted graphs, dynamic settings with node arrivals and departures, and theoretical extensions to time varying degree-corrected stochastic block models (Karrer & Newman, 2011).

## Acknowledgements

We thank Yuki Takazawa, Ryoma Kondo, Kohei Miyaguchi, Tilmann Altwicker and Zhivko Taushanov for helpful discussions. R.H. is supported by JST FOREST Program (JP-MJFR216Q), JST PRESTO Program (JPMJPR2469), Grant-in-Aid for Scientific Research (KAKENHI, JP24K03043), JSPS International Joint Research Program (JRPs with SNSF: 20251501), and the UTEC-UTokyo FSI Research Grant Program.

## Impact Statement

This paper presents work whose goal is to advance the field of machine learning through improved methods for analyzing dynamic graphs. The proposed techniques are general purpose and are not tied to any specific application involving human subjects or sensitive data. As such, we do not identify any unique ethical concerns or foreseeable negative societal impacts arising specifically from this work.

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

# A. Appendix

This appendix contains additional theoretical analysis, counterexamples, proofs, and experimental details that support the main paper.

# B. Notation Summary

Table 2 summarizes the notation used throughout the paper. Superscripts $(t)$ always index time and are written explicitly when a quantity varies across snapshots. In particular, node level community labels $\mathbf{z}_i^{(t)}$ may vary with $t$, while the matrix $\mathcal{Z}$ denotes equivalence classes of nodes with identical population connectivity profiles and is *not* time indexed. Tildes $\tilde{\cdot}$ denote population (noise-free) quantities. All matrices are written in boldface for clarity.

| Symbol | Meaning |
|---|---|
| **Indices and dimensions** | |
| $n, T$ | Number of nodes; number of time steps. |
| $i, j \in \{1, \ldots, n\}$ | Node indices. |
| $s, t \in \{1, \ldots, T\}$ | Time indices. |
| $K$ | Number of latent communities / equivalence classes. |
| $d$ | Embedding dimension. |
| **Dynamic graphs and observed quantities** | |
| $\mathcal{G} = \{G^{(1)}, \ldots, G^{(T)}\}$ | Dynamic graph as a sequence of snapshots. |
| $\mathbf{A}^{(t)} \in \{0, 1\}^{n \times n}$ | Adjacency matrix at time $t$. |
| $\mathcal{A} = [\mathbf{A}^{(1)} | \cdots | \mathbf{A}^{(T)}]$ | Unfolded adjacency matrix. |
| $\mathbf{d}^{(t)}, \mathbf{d}_i^{(t)}$ | Degree vector at time $t$; degree of node $i$. |
| $\mathbf{D}^{(t)} = \mathrm{diag}(\mathbf{d}^{(t)})$ | Degree matrix at time $t$. |
| $\mathbf{d}^{(1:T)}, \mathbf{D}^{(1:T)}$ | Aggregated degrees across time. |
| **Population (noise-free) quantities** | |
| $\mathbf{P}^{(t)} \in [0, 1]^{n \times n}$ | Edge probability matrix at time $t$. |
| $\tilde{\mathbf{d}}^{(t)}, \tilde{\mathbf{d}}_i^{(t)}$ | Population degree vector; population degree of node $i$. |
| $\tilde{\mathbf{D}}^{(t)} = \mathrm{diag}(\tilde{\mathbf{d}}^{(t)})$ | Population degree matrix at time $t$. |
| **Dynamic stochastic block model (DSBM)** | |
| $\mathbf{z}^{(t)} \in \{1, \ldots, K\}^n$ | Community assignment vector at time $t$. |
| $\boldsymbol{\pi} \in \mathbb{R}^K$ | Marginal community proportion vector. |
| $\mathbf{n}^{(t)} \in \mathbb{N}^K$ | Community size vector at time $t$ (used only in the generative model). |
| $\rho$ | Sparsity parameter. |
| $\mathbf{B}^{(t)} \in [0, 1]^{K \times K}$ | Block probability matrix at time $t$. |
| $\mathbf{P}_{ij}^{(t)} = \rho \mathbf{B}_{\mathbf{z}_i^{(t)} \mathbf{z}_j^{(t)}}^{(t)}$ | DSBM edge probability. |
| **Equivalence classes and block structure** | |
| $\mathcal{Z} \in \{0, 1\}^{n \times K}$ | Indicator matrix for equivalence classes of nodes with identical population connectivity profiles. |
| $\mathcal{S} = \mathrm{span}(\mathcal{Z})$ | Subspace spanned by the columns of the equivalence-class indicator matrix $\mathcal{Z}$. |
| $\mathbf{n}, \mathbf{N}$ | Equivalence class size vector and diagonal matrix, with $\mathbf{N} = \mathrm{diag}(\mathbf{n})$. |
| **ULSE-n1 operators** | |
| $\mathbf{L}_{\mathrm{n1}}^{(t)} = \mathbf{I} - \mathbf{D}^{(t)-1/2} \mathbf{A}^{(t)} \mathbf{D}^{(t)-1/2}$ | Normalized Laplacian at time $t$. |
| $\mathcal{L}_{\mathrm{n1}}, \mathcal{L}_{\mathrm{n1}}^{-t}$ | Unfolded normalized Laplacian; with block $t$ removed. |
| $\tilde{\mathbf{L}}_{\mathrm{n1}}^{(t)}, \tilde{\mathcal{L}}_{\mathrm{n1}}$ | Population counterparts. |
| $\bar{\mathcal{L}}_{\mathrm{n1}}$ | community level population ULSE-n1 operator. |
| **ULSE-n2 operators** | |
| $\mathbf{L}_{\mathrm{n2}}^{(t)} = -\mathbf{D}^{(1:T)-1/2} \mathbf{A}^{(t)} \mathbf{D}^{(t)-1/2}$ | ULSE-n2 operator at time $t$. |
| $\mathcal{L}_{\mathrm{n2}}$ | Unfolded ULSE-n2 operator. |
| $\tilde{\mathbf{L}}_{\mathrm{n2}}^{(t)}, \tilde{\mathcal{L}}_{\mathrm{n2}}$ | Population counterparts. |
| $\bar{\mathcal{L}}_{\mathrm{n2}}$ | community level population ULSE-n2 operator. |
| **Spectral quantities** | |
| $\sigma_k, \tilde{\sigma}_k$ | $k$-th smallest nontrivial singular value of an unfolded Laplacian; empirical and population versions. |
| $\bar{\lambda}_1 \leq \cdots \leq \bar{\lambda}_K$ | Nontrivial singular values of a community level population operator. |
| **Conductance and Cheeger-type quantities** | |
| $\phi_k(\mathcal{G}), \phi_k(G^{(t)})$ | $k$-way conductance of the dynamic graph; of snapshot $t$. |
| **Embeddings and SVD conventions** | |
| $\hat{\mathbf{X}}, \hat{\mathbf{Y}}^{(t)}$ | Anchor embedding; dynamic embedding at time $t$. |
| $\tilde{\mathbf{X}}, \tilde{\mathbf{Y}}^{(t)}$ | Population embeddings. |
| $\mathbf{W} \in \mathbb{O}(d \times d), \mathbb{O}(n \times d)$ | Orthogonal matrices and Stiefel manifolds (orthonormal columns). |
| **Norms and asymptotics** | |
| $\|\cdot\|_2, \|\cdot\|_F$ | Spectral and Frobenius norms. |
| $\mathcal{O}(\cdot), \Omega(\cdot), \omega(\cdot)$ | Asymptotic notation (a.s. when stated). |
| $\mathrm{poly}(k)$ | Polynomial in $k$ with fixed degree. |

*Table 2.* Summary of notation.

## C. Limitations of Context-Aware Perturbation Embedding

Zhao et al. (2021) propose a context-aware spectral embedding framework for dynamic graphs, in which each snapshot $G^{(t)}$ is modeled as a perturbation of a shared global context graph $G^{\mathrm{ctx}}$. Let $\mathbf{A}^{(t)}$ denote the adjacency matrix at time $t$, and define the context adjacency matrix by entrywise averaging

$$\mathbf{A}^{\mathrm{ctx}}_{ij} = \frac{1}{T} \sum_{t=1}^{T} \mathbf{A}^{(t)}_{ij}.$$

Let $\mathbf{D}^{\mathrm{ctx}}$ be the associated degree matrix, and define the normalized Laplacian of the context graph as

$$\mathbf{L}^{\mathrm{ctx}} = \mathbf{I} - (\mathbf{D}^{\mathrm{ctx}})^{-1/2} \mathbf{A}^{\mathrm{ctx}} (\mathbf{D}^{\mathrm{ctx}})^{-1/2}.$$

Each snapshot Laplacian is written as

$$\mathbf{L}^{(t)}_{\mathrm{snap}} = \mathbf{L}^{\mathrm{ctx}} + \mathbf{\Delta}^{(t)},$$

where $\mathbf{\Delta}^{(t)}$ is assumed to be a small symmetric perturbation. Suppose that $\mathbf{L}^{\mathrm{ctx}}$ has a nontrivial spectral gap $\delta^{\mathrm{ctx}} > 0$ and that the perturbations satisfy

$$\varepsilon := \max_{t \in \{1, \dots, T\}} \|\mathbf{\Delta}^{(t)}\|_2 < \infty.$$

Standard matrix perturbation theory (Stewart & Sun, 1990) yields first-order approximations for the eigenvalues and eigenvectors of $\mathbf{L}^{(t)}_{\mathrm{snap}}$. Let $\lambda^{\mathrm{ctx}}_i$ and $\mathbf{u}^{\mathrm{ctx}}_i$ denote the eigenpairs of $\mathbf{L}^{\mathrm{ctx}}$. For each $i$,

$$\lambda^{(t)}_i = \lambda^{\mathrm{ctx}}_i + (\mathbf{u}^{\mathrm{ctx}}_i)^\top \mathbf{\Delta}^{(t)} \mathbf{u}^{\mathrm{ctx}}_i + o(\varepsilon), \tag{1}$$

$$\mathbf{u}^{(t)}_i = \mathbf{u}^{\mathrm{ctx}}_i + \sum_{j \neq i} \frac{(\mathbf{u}^{\mathrm{ctx}}_j)^\top \mathbf{\Delta}^{(t)} \mathbf{u}^{\mathrm{ctx}}_i}{\lambda^{\mathrm{ctx}}_i - \lambda^{\mathrm{ctx}}_j} \mathbf{u}^{\mathrm{ctx}}_j + o(\varepsilon). \tag{2}$$

These expansions suggest constructing the embedding at time $t$ using the perturbed eigenvectors $\mathbf{u}^{(t)}_1, \dots, \mathbf{u}^{(t)}_k$. However, even when stability notions are relaxed to allow time varying communities or time varying connectivity profiles, this perturbative construction does not induce an injective map from graph snapshots to embeddings: structurally distinct snapshot Laplacians may yield identical collections of perturbed eigenvectors (up to the usual orthogonal indeterminacy). Consequently, the context-aware perturbation framework cannot, in general, guarantee cross-sectional stability or longitudinal stability.

This limitation appears explicitly in the proof of Theorem 4.1 of Zhao et al. (2021), which claims that

$$\mathbf{U}^{(s)}_{\mathrm{ctx}} = \mathbf{U}^{(t)}_{\mathrm{ctx}} \quad \Rightarrow \quad \mathbf{L}^{(s)}_{\mathrm{snap}} = \mathbf{L}^{(t)}_{\mathrm{snap}},$$

where $\mathbf{U}^{(t)}_{\mathrm{ctx}}$ denotes the matrix whose columns are the embedding eigenvectors at time $t$. The argument relies on the condition

$$(\mathbf{u}^{\mathrm{ctx}}_i)^\top (\mathbf{L}^{(t)}_{\mathrm{snap}} - \mathbf{L}^{(s)}_{\mathrm{snap}}) \mathbf{u}^{\mathrm{ctx}}_i = 0 \quad \text{for all } i,$$

but this condition is insufficient to conclude equality of the two Laplacians. The vectors $\{\mathbf{u}^{\mathrm{ctx}}_i\}$ do not determine a symmetric matrix uniquely: there may exist nonzero symmetric components of $\mathbf{L}^{(t)}_{\mathrm{snap}} - \mathbf{L}^{(s)}_{\mathrm{snap}}$ that are orthogonal to all rank-one projectors $\mathbf{u}^{\mathrm{ctx}}_i \mathbf{u}^{\mathrm{ctx}\top}_i$. Therefore, equality of the embedding eigenvectors (or their span) does not imply equality of the underlying snapshot Laplacians, and the claimed injectivity step fails.

### C.1. Counterexample

To illustrate the failure of injectivity in the context-aware perturbation framework, we construct two non-isomorphic graphs whose context-aware spectral embeddings coincide. Consider two three-node graphs $G^{(1)}$ and $G^{(2)}$, shown in Figure 2, with normalized snapshot Laplacians

$$\mathbf{L}^{(1)}_{\mathrm{snap}} = \begin{bmatrix} 1 & 0 & -\sqrt{1/2} \\ 0 & 1 & -\sqrt{1/2} \\ -\sqrt{1/2} & -\sqrt{1/2} & 1 \end{bmatrix}, \quad \mathbf{L}^{(2)}_{\mathrm{snap}} = \begin{bmatrix} 1 & -\sqrt{1/2} & 0 \\ -\sqrt{1/2} & 1 & -\sqrt{1/2} \\ 0 & -\sqrt{1/2} & 1 \end{bmatrix}. \tag{3}$$

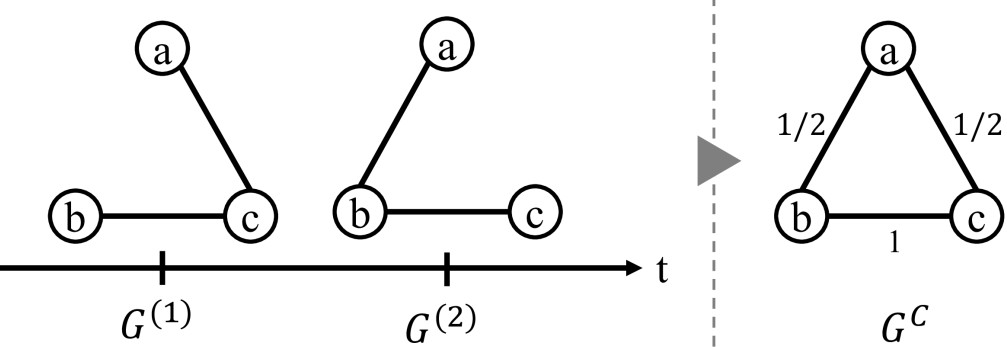

*Figure 2.* Counterexample demonstrating the failure of Theorem 4.1 in Zhao et al. (2021). Despite structural differences between $G^{(1)}$ and $G^{(2)}$, their spectral embeddings under the context-aware perturbation model are identical.

These graphs are not isomorphic, as their adjacency patterns differ.

The context Laplacian is defined as the entrywise average of the snapshot Laplacians,

$$\mathbf{L}_{\text{ctx}} = \frac{1}{2}\left(\mathbf{L}_{\text{snap}}^{(1)} + \mathbf{L}_{\text{snap}}^{(2)}\right) = \begin{bmatrix} 1 & -\sqrt{1/6} & -\sqrt{1/6} \\ -\sqrt{1/6} & 1 & -2/3 \\ -\sqrt{1/6} & -2/3 & 1 \end{bmatrix}.$$

Its eigendecomposition yields the eigenpairs $\lambda_1^{\text{ctx}} = 0$ with $\mathbf{u}_1^{\text{ctx}} = (\frac{1}{2}, \sqrt{\frac{3}{8}}, \sqrt{\frac{3}{8}})^\top$, $\lambda_2^{\text{ctx}} = \frac{4}{3}$ with $\mathbf{u}_2^{\text{ctx}} = (-\frac{\sqrt{3}}{2}, \sqrt{\frac{1}{8}}, \sqrt{\frac{1}{8}})^\top$, and $\lambda_3^{\text{ctx}} = \frac{5}{3}$ with $\mathbf{u}_3^{\text{ctx}} = (0, -\frac{1}{\sqrt{2}}, \frac{1}{\sqrt{2}})^\top$.

Define the perturbation matrices $\boldsymbol{\Delta}^{(t)} = \mathbf{L}_{\text{snap}}^{(t)} - \mathbf{L}_{\text{ctx}}$. Explicitly,

$$\boldsymbol{\Delta}^{(1)} = \begin{bmatrix} 0 & \sqrt{1/6} & \sqrt{1/6} - \sqrt{1/2} \\ \sqrt{1/6} & 0 & 2/3 - \sqrt{1/2} \\ \sqrt{1/6} - \sqrt{1/2} & 2/3 - \sqrt{1/2} & 0 \end{bmatrix}, \tag{4}$$

$$\boldsymbol{\Delta}^{(2)} = \begin{bmatrix} 0 & \sqrt{1/6} - \sqrt{1/2} & \sqrt{1/6} \\ \sqrt{1/6} - \sqrt{1/2} & 0 & 2/3 - \sqrt{1/2} \\ \sqrt{1/6} & 2/3 - \sqrt{1/2} & 0 \end{bmatrix}. \tag{5}$$

We apply the first-order eigenvalue perturbation formula

$$\Delta\lambda_i^{(t)} = \mathbf{u}_i^{\text{ctx}\top}\boldsymbol{\Delta}^{(t)}\mathbf{u}_i^{\text{ctx}}, \qquad i = 1, 2, 3. \tag{6}$$

Substituting $\boldsymbol{\Delta}^{(1)}$ into (6) and evaluating the quadratic forms yields

$$(\Delta\lambda_1^{(1)}, \Delta\lambda_2^{(1)}, \Delta\lambda_3^{(1)}) = \left(1 - \tfrac{3\sqrt{2}}{8} - \tfrac{\sqrt{3}}{4},\ \tfrac{\sqrt{3}}{4} - \tfrac{1}{3} - \tfrac{\sqrt{2}}{8},\ \tfrac{\sqrt{2}}{2} - \tfrac{2}{3}\right).$$

Repeating the same computation with $\boldsymbol{\Delta}^{(2)}$ gives

$$(\Delta\lambda_1^{(2)}, \Delta\lambda_2^{(2)}, \Delta\lambda_3^{(2)}) = \left(1 - \tfrac{3\sqrt{2}}{8} - \tfrac{\sqrt{3}}{4},\ \tfrac{\sqrt{3}}{4} - \tfrac{1}{3} - \tfrac{\sqrt{2}}{8},\ \tfrac{\sqrt{2}}{2} - \tfrac{2}{3}\right).$$

Thus, the first-order eigenvalue shifts coincide for $t = 1$ and $t = 2$.

An analogous substitution into the first-order eigenvector perturbation formula

$$\Delta\mathbf{u}_i^{(t)} = \sum_{j\neq i} \frac{\mathbf{u}_j^{\text{ctx}\top}\boldsymbol{\Delta}^{(t)}\mathbf{u}_i^{\text{ctx}}}{\lambda_i^{\text{ctx}} - \lambda_j^{\text{ctx}}}\ \mathbf{u}_j^{\text{ctx}} \tag{7}$$

shows that the first-order eigenvector corrections are also identical for $t = 1$ and $t = 2$. Consequently, the perturbed eigenvalue matrices coincide, $\boldsymbol{\Lambda}^{(1)} = \boldsymbol{\Lambda}^{(2)}$, and the corresponding context-aware embedding eigenvector matrices satisfy $\mathbf{U}_{\text{ctx}}^{(1)} = \mathbf{U}_{\text{ctx}}^{(2)}$.

Therefore, under the context-aware perturbation model, two non-isomorphic graphs produce identical spectral embeddings. This contradicts the injectivity claim of Theorem 4.1 in Zhao et al. (2021) and demonstrates that the perturbative construction fails to uniquely encode snapshot level graph structure, even in this minimal three-node example.

## D. On Theorem 5 of Davis et al. (2023)

Theorem 5 of Davis et al. (2023) claims that the dynamic embeddings $\hat{\mathbf{Y}}$ produced by the dilated unfolded embedding are both cross-sectional (spatially) and longitudinal (temporally) stable. We show that the key permutation argument used to justify this claim fails: the required permutation matrix does not, in general, map the class of dilated unfolded adjacency matrices to itself. Consequently, the asserted invariance (and hence the theorem) does not hold in full generality.

Let $\mathcal{A} \in \{0, 1\}^{(n+nT)\times(n+nT)}$ denote the dilated unfolded adjacency matrix, and let $\mathbb{A}$ denote the set of all matrices with this dilated unfolded structure. Davis et al. (2023) argue that for a permutation matrix $\boldsymbol{\Pi}$ that swaps node $i$ at time $u$ with node $j$ at time $t$, one has

$$\sum_{\mathbf{a}\in\mathbb{A}} \Pr\left(\hat{\mathbf{Y}}_{i:}^{(s)} = v_1,\ \hat{\mathbf{Y}}_{j:}^{(t)} = v_2 \,\Big|\, \mathcal{A} = \mathbf{a}\right)\Pr(\mathcal{A} = \mathbf{a})$$

$$= \sum_{\mathbf{a}\in\mathbb{A}} \Pr\left(\hat{\mathbf{Y}}_{i:}^{(s)} = v_2,\ \hat{\mathbf{Y}}_{j:}^{(t)} = v_1 \,\Big|\, \mathcal{A} = \boldsymbol{\Pi}\mathbf{a}\boldsymbol{\Pi}^\top\right)\Pr\left(\mathcal{A} = \boldsymbol{\Pi}\mathbf{a}\boldsymbol{\Pi}^\top\right).$$

For this identity to be valid, it is necessary that $\boldsymbol{\Pi}\mathbf{a}\boldsymbol{\Pi}^\top \in \mathbb{A}$ whenever $\mathbf{a} \in \mathbb{A}$, i.e., that conjugation by $\boldsymbol{\Pi}$ preserves the dilated unfolded structure.

However, the permutation $\boldsymbol{\Pi}$ required by the argument must swap the embedding of node $i$ at time $u$ with that of node $j$ at time $t$, while leaving all other components unchanged, i.e.,

$$\boldsymbol{\Pi}\begin{bmatrix}\hat{\mathbf{X}} \\ \hat{\mathbf{Y}}_{1:}^{(1)} \\ \vdots \\ \hat{\mathbf{Y}}_{i:}^{(s)} \\ \vdots \\ \hat{\mathbf{Y}}_{j:}^{(t)} \\ \vdots \\ \hat{\mathbf{Y}}_{n:}^{(t)}\end{bmatrix} = \begin{bmatrix}\hat{\mathbf{X}} \\ \hat{\mathbf{Y}}_{1:}^{(1)} \\ \vdots \\ \hat{\mathbf{Y}}_{j:}^{(t)} \\ \vdots \\ \hat{\mathbf{Y}}_{i:}^{(s)} \\ \vdots \\ \hat{\mathbf{Y}}_{n:}^{(t)}\end{bmatrix}.$$

Such a permutation necessarily places ones at indices $(nt + i, nu + j)$ and $(nu + j, nt + i)$, has identity entries elsewhere except at these indices, and zeros in all remaining positions. But conjugating a dilated unfolded adjacency matrix $\mathbf{a} \in \mathbb{A}$ by this $\mathbf{\Pi}$ generally destroys the dilation pattern (i.e., the prescribed block placement of within-time and cross-time entries). Therefore $\mathbf{\Pi}\mathbf{a}\mathbf{\Pi}^\top$ need not belong to $\mathbb{A}$, and the set invariance claim

$$\{\mathbf{a} \in \mathbb{A} : \mathbf{\Pi}\mathbf{a}\mathbf{\Pi}^\top \in \mathbb{A}\} = \mathbb{A}$$

is false in general. The equality above is thus not justified, and the subsequent stability conclusion does not follow.

This structural failure yields violations of both spatial and temporal stability. For spatial stability, one may take

$$\mathbf{P}^{(1)} = \begin{bmatrix} 0.3 & 0.3 \\ 0.3 & 0.3 \end{bmatrix}, \qquad \mathbf{P}^{(2)} = \begin{bmatrix} 1 & 1 \\ 1 & 0 \end{bmatrix}.$$

For temporal stability, a counterexample is given by

$$\mathbf{P}^{(1)} = \begin{bmatrix} 1 & 0.3 \\ 0.3 & 1 \end{bmatrix}, \qquad \mathbf{P}^{(2)} = \begin{bmatrix} 1 & 0.3 \\ 0.3 & 0 \end{bmatrix}.$$

In both cases, the dilated unfolded embedding fails to satisfy the corresponding stability condition, contradicting the claim of Theorem 5.

## E. Proof of Theorem 4.2

**Lemma E.1** (Lower bound for the minimum degree). *For each $t \in \{1, \ldots, T\}$, the minimum population and sample degrees satisfy*

$$\tilde{\mathbf{d}}_{\min}^{(t)} := \min_{i \in [N]} \tilde{\mathbf{d}}_i^{(t)} = \Omega(\rho n) \quad a.s., \qquad \mathbf{d}_{\min}^{(t)} := \min_{i \in [N]} \mathbf{d}_i^{(t)} = \Omega(\rho n) \quad a.s.$$

*Proof.* We treat the population degrees and the sample degrees separately.

Since $\tilde{\mathbf{d}}^{(t)} = \mathbf{P}^{(t)}\mathbf{1}$, we have $\tilde{\mathbf{d}}_i^{(t)} = \sum_{j=1}^{n} \mathbf{P}_{ij}^{(t)}$. Under the DSBM, $\mathbf{P}_{ij}^{(t)} = \rho \mathbf{B}_{\mathbf{z}_i^{(t)} \mathbf{z}_j^{(t)}}^{(t)}$, and therefore, for every $i$,

$$\tilde{\mathbf{d}}_i^{(t)} = \rho \sum_{j=1}^{n} \mathbf{B}_{\mathbf{z}_i^{(t)} \mathbf{z}_j^{(t)}}^{(t)} \geq \rho n \, \mathbf{B}_{\min}^{(t)}.$$

Hence $\tilde{\mathbf{d}}_{\min}^{(t)} \geq \rho n \, \mathbf{B}_{\min}^{(t)}$, and since $\mathbf{B}_{\min}^{(t)} > 0$ by assumption, it follows that $\tilde{\mathbf{d}}_{\min}^{(t)} = \Omega(\rho n)$ (deterministically, hence a.s.).

Fix $i$. Because $\mathbf{d}_i^{(t)} = \sum_{j=1}^{n} \mathbf{A}_{ij}^{(t)}$ and $\mathbf{A}_{ij}^{(t)} \sim \text{Bernoulli}(\mathbf{P}_{ij}^{(t)})$ are independent conditional on $\mathbf{P}^{(t)}$, $\mathbb{E}[\mathbf{d}_i^{(t)}] = \tilde{\mathbf{d}}_i^{(t)}$. Moreover,

$$\text{Var}(\mathbf{A}_{ij}^{(t)}) = \mathbf{P}_{ij}^{(t)}(1 - \mathbf{P}_{ij}^{(t)}) \leq \mathbf{P}_{ij}^{(t)} \leq \rho,$$

so

$$\sum_{j=1}^{n} \text{Var}(\mathbf{A}_{ij}^{(t)}) \leq n\rho.$$

Bernstein's inequality implies that for any $s > 0$,

$$\Pr\left( \left| \mathbf{d}_i^{(t)} - \tilde{\mathbf{d}}_i^{(t)} \right| \geq s \right) \leq 2 \exp\left( -\frac{s^2/2}{n\rho + s/3} \right).$$

Applying a union bound over $i \in \{1, \ldots, n\}$ yields

$$\Pr\left( \max_i \left| \mathbf{d}_i^{(t)} - \tilde{\mathbf{d}}_i^{(t)} \right| \geq s \right) \leq 2n \exp\left( -\frac{s^2/2}{n\rho + s/3} \right).$$

Fix $\alpha > 0$. To guarantee that the right-hand side is at most $n^{-\alpha}$, it suffices to choose $s$ such that

$$2n \exp\left( -\frac{s^2/2}{n\rho + s/3} \right) \leq n^{-\alpha}.$$

Taking logarithms and rearranging yields

$$\frac{s^2}{2\,(n\rho + s/3)} \geq (\alpha + 1)\log n + \log 2.$$

Since $\rho = \omega(\log n/n)$, one can verify that

$$s = \Theta\left(\rho^{1/2} n^{1/2} \log^{1/2} n\right)$$

satisfies this inequality. Moreover, since $\rho = \omega(\log n/n)$, we also have

$$\sqrt{n\rho \log n} = o(\rho n),$$

and thus

$$\max_i \left|\mathbf{d}_i^{(t)} - \tilde{\mathbf{d}}_i^{(t)}\right| = o(\rho n) \quad \text{a.s.}$$

Combining this with $\tilde{\mathbf{d}}_{\min}^{(t)} = \Omega(\rho n)$ gives

$$\mathbf{d}_{\min}^{(t)} \geq \tilde{\mathbf{d}}_{\min}^{(t)} - \max_i \left|\mathbf{d}_i^{(t)} - \tilde{\mathbf{d}}_i^{(t)}\right| = \Omega(\rho n) \quad \text{a.s.}$$

$\square$

By Lemma E.1, the degree matrices $\tilde{\mathbf{D}}^{(t)}$ and $\mathbf{D}^{(t)}$ are almost surely invertible, so the unfolded normalized Laplacians $\tilde{\mathcal{L}}_{n1}$ and $\mathcal{L}_{n1}$ are well defined.

**Lemma E.2** (Deviation bound for the unfolded normalized Laplacian).

$$\|\mathcal{L}_{n1} - \tilde{\mathcal{L}}_{n1}\|_2 = \mathcal{O}\left((\rho n)^{-1/2}\right) \quad \text{a.s.}$$

*Proof.* Write $\mathcal{L}_{n1} = [\mathbf{L}_{n1}^{(1)}|\cdots|\mathbf{L}_{n1}^{(T)}]$ and $\tilde{\mathcal{L}}_{n1} = [\tilde{\mathbf{L}}_{n1}^{(1)}|\cdots|\tilde{\mathbf{L}}_{n1}^{(T)}]$. Then

$$\mathcal{L}_{n1} - \tilde{\mathcal{L}}_{n1} = \left[\mathbf{L}_{n1}^{(1)} - \tilde{\mathbf{L}}_{n1}^{(1)} \mid \cdots \mid \mathbf{L}_{n1}^{(T)} - \tilde{\mathbf{L}}_{n1}^{(T)}\right].$$

Using the triangle inequality for spectral norm,

$$\|\mathcal{L}_{n1} - \tilde{\mathcal{L}}_{n1}\|_2 \leq \sum_{t=1}^{T} \|\mathbf{L}_{n1}^{(t)} - \tilde{\mathbf{L}}_{n1}^{(t)}\|_2,$$

so it suffices to control $\|\mathbf{L}_{n1}^{(t)} - \tilde{\mathbf{L}}_{n1}^{(t)}\|_2$ uniformly over $t$.

Fix $t$. Under the DSBM, $\max_{i,j} \mathbf{P}_{ij}^{(t)} \leq \rho$, and by Lemma E.1,

$$\min(\mathbf{d}_{\min}^{(t)}, \tilde{\mathbf{d}}_{\min}^{(t)}) = \Omega(\rho n) \quad \text{a.s.}$$

Moreover, since $\mathbf{B}_{\min}^{(t)} > 0$ and $\rho = \omega(\log n/n)$, there exist constants $N, c_0 > 0$ such that for all $n \geq N$,

$$n \max_{i,j} \mathbf{P}_{ij}^{(t)} \geq n\rho\, \mathbf{B}_{\min}^{(t)} \geq c_0 \log n.$$

Therefore, Theorem 3.1 of Deng et al. (2021) applies and yields

$$\|\mathbf{L}_{n1}^{(t)} - \tilde{\mathbf{L}}_{n1}^{(t)}\|_2 = \mathcal{O}\left(\frac{(n \max_{i,j} \mathbf{P}_{ij}^{(t)})^{5/2}}{\min(\mathbf{d}_{\min}^{(t)}, \tilde{\mathbf{d}}_{\min}^{(t)})^3}\right) \quad \text{a.s.}$$

Substituting $n \max_{i,j} \mathbf{P}_{ij}^{(t)} \leq n\rho$ and $\min(\mathbf{d}_{\min}^{(t)}, \tilde{\mathbf{d}}_{\min}^{(t)}) = \Omega(\rho n)$ gives

$$\|\mathbf{L}_{n1}^{(t)} - \tilde{\mathbf{L}}_{n1}^{(t)}\|_2 = \mathcal{O}\left(\frac{(n\rho)^{5/2}}{(\rho n)^3}\right) = \mathcal{O}\left((\rho n)^{-1/2}\right) \quad \text{a.s.}$$

Finally, combining over $t = 1, \ldots, T$ yields

$$\|\mathcal{L}_{n1} - \tilde{\mathcal{L}}_{n1}\|_2 \leq \sum_{t=1}^{T} \|\mathbf{L}_{n1}^{(t)} - \tilde{\mathbf{L}}_{n1}^{(t)}\|_2 = \mathcal{O}\left((\rho n)^{-1/2}\right) \quad \text{a.s.,}$$

absorbing the (fixed) factor $T$ into the $\mathcal{O}(\cdot)$ term. $\qquad \square$

**Lemma E.3** (SVD of the noise-free unfolded normalized Laplacian). *Assume* $\mathbf{N} = \mathrm{diag}(\mathbf{n})$ *is invertible and let* $\mathcal{S} = \mathrm{span}(\mathcal{Z})$. *The unfolded population operator* $\tilde{\mathcal{L}}_{n1} = [\tilde{\mathbf{L}}_{n1}^{(1)} | \cdots | \tilde{\mathbf{L}}_{n1}^{(T)}]$ *has the following singular-value structure:*

- *A singular value* $\sqrt{T}$ *with multiplicity* $n - K$, *whose left singular space is* $\mathcal{S}^{\perp}$.

- *The remaining* $K$ *singular values are* $\tilde{\lambda}_1, \ldots, \tilde{\lambda}_K$. *Their left singular vectors lie in* $\mathcal{S}$ *and can be written as* $\tilde{\mathbf{u}}^{(i)} = \mathcal{Z}\mathbf{N}^{-1/2}\tilde{\mathbf{x}}^{(i)}$, *where* $\{\tilde{\mathbf{x}}^{(i)}\}_{i=1}^{K}$ *is an orthonormal set in* $\mathbb{R}^K$.

*Proof.* We decompose $\mathbb{R}^n = \mathcal{S} \oplus \mathcal{S}^{\perp}$.

**Step 1: singular vectors in $\mathcal{S}^{\perp}$** Let $\{\mathbf{x}^{(1)}, \ldots, \mathbf{x}^{(n-K)}\}$ be an orthonormal basis of $\mathcal{S}^{\perp}$. Fix $t \in \{1, \ldots, T\}$. Recall that

$$\tilde{\mathbf{L}}_{n1}^{(t)} = \mathbf{I} - \tilde{\mathbf{D}}^{(t)-1/2}\mathbf{P}^{(t)}\tilde{\mathbf{D}}^{(t)-1/2}.$$

For $\mathbf{x}^{(i)} \in \mathcal{S}^{\perp}$, the $r$th coordinate of $\tilde{\mathbf{D}}^{(t)-1/2}\mathbf{P}^{(t)}\tilde{\mathbf{D}}^{(t)-1/2}\mathbf{x}^{(i)}$ is

$$\sum_{j=1}^{n} \frac{\mathbf{P}_{rj}^{(t)}}{\sqrt{\tilde{d}_r^{(t)}\tilde{d}_j^{(t)}}} \mathbf{x}_j^{(i)}$$

$$= \sum_{k=1}^{K} \sum_{j:\mathbf{z}_j=k} \frac{\rho \mathbf{B}_{\mathbf{z}_r^{(t)}c^{(t)}(k)}^{(t)}}{\sqrt{\left(\sum_{m=1}^{K} n_m \rho \mathbf{B}_{\mathbf{z}_r^{(t)}c^{(t)}(m)}^{(t)}\right)\left(\sum_{m=1}^{K} n_m \rho \mathbf{B}_{c^{(t)}(k)c^{(t)}(m)}^{(t)}\right)}} \mathbf{x}_j^{(i)}$$

$$= \sum_{k=1}^{K} \frac{\mathbf{B}_{\mathbf{z}_r^{(t)}c^{(t)}(k)}^{(t)}}{\sqrt{\left(\sum_{m=1}^{K} n_m \mathbf{B}_{\mathbf{z}_r^{(t)}c^{(t)}(m)}^{(t)}\right)\left(\sum_{m=1}^{K} n_m \mathbf{B}_{c^{(t)}(k)c^{(t)}(m)}^{(t)}\right)}} \sum_{j:\mathbf{z}_j=k} \mathbf{x}_j^{(i)}.$$

Since $\mathbf{x}^{(i)} \in \mathcal{S}^{\perp}$, we have $\sum_{j:\mathbf{z}_j=k} \mathbf{x}_j^{(i)} = 0$ for every $k$, and therefore

$$\tilde{\mathbf{D}}^{(t)-1/2}\mathbf{P}^{(t)}\tilde{\mathbf{D}}^{(t)-1/2}\mathbf{x}^{(i)} = \mathbf{0}.$$

Hence

$$\tilde{\mathbf{L}}_{n1}^{(t)}\mathbf{x}^{(i)} = \mathbf{x}^{(i)}.$$

Since $\tilde{\mathbf{L}}_{n1}^{(t)}$ is symmetric,

$$\tilde{\mathbf{L}}_{n1}^{(t)}\tilde{\mathbf{L}}_{n1}^{(t)\top}\mathbf{x}^{(i)} = \mathbf{x}^{(i)}.$$

Summing over $t$ yields

$$\tilde{\mathcal{L}}_{n1}\tilde{\mathcal{L}}_{n1}^{\top}\mathbf{x}^{(i)} = \sum_{t=1}^{T} \mathbf{x}^{(i)} = T\mathbf{x}^{(i)}.$$

Thus each $\mathbf{x}^{(i)}$ is a left singular vector with singular value $\sqrt{T}$. Since $\dim(\mathcal{S}^{\perp}) = n - K$, this singular value has multiplicity $n - K$.

**Step 2: singular vectors in $\mathcal{S}$.** Let $\{\tilde{\mathbf{x}}^{(1)}, \ldots, \tilde{\mathbf{x}}^{(K)}\}$ be an orthonormal basis of $\mathbb{R}^K$ and define

$$\tilde{\mathbf{u}}^{(i)} = \mathcal{Z}\mathbf{N}^{-1/2}\tilde{\mathbf{x}}^{(i)}.$$

Using $\mathcal{Z}^{\top}\mathcal{Z} = \mathbf{N}$, we obtain

$$\tilde{\mathbf{u}}^{(i)\top}\tilde{\mathbf{u}}^{(j)} = \delta_{ij},$$

so $\{\tilde{\mathbf{u}}^{(i)}\}_{i=1}^{K}$ is an orthonormal basis of $\mathcal{S}$.

Fix $t$. The $(r, \ell)$ entry of $\tilde{\mathbf{D}}^{(t)-1/2}\mathbf{P}^{(t)}\tilde{\mathbf{D}}^{(t)-1/2}\mathcal{Z}$ is

$$\sum_{j=1}^{n} \frac{\mathbf{P}_{rj}^{(t)}}{\sqrt{\tilde{d}_r^{(t)}\tilde{d}_j^{(t)}}} \mathcal{Z}_{j\ell} = \sum_{j:\mathbf{z}_j=\ell} \frac{\rho\,\mathbf{B}_{\mathbf{z}_r^{(t)}c^{(t)}(\ell)}^{(t)}}{\sqrt{\left(\sum_{m=1}^{K} n_m \rho\,\mathbf{B}_{\mathbf{z}_r^{(t)}c^{(t)}(m)}^{(t)}\right)\left(\sum_{m=1}^{K} n_m \rho\,\mathbf{B}_{c^{(t)}(\ell)c^{(t)}(m)}^{(t)}\right)}}$$

$$= \frac{n_\ell\,\mathbf{B}_{\mathbf{z}_r^{(t)}c^{(t)}(\ell)}^{(t)}}{\sqrt{\left(\sum_{m=1}^{K} n_m \mathbf{B}_{\mathbf{z}_r^{(t)}c^{(t)}(m)}^{(t)}\right)\left(\sum_{m=1}^{K} n_m \mathbf{B}_{c^{(t)}(\ell)c^{(t)}(m)}^{(t)}\right)}}.$$

Comparing with the $(k, \ell)$ entry of $\mathbf{I} - \mathbf{N}^{-1/2}\tilde{\mathbf{M}}_{n1}^{(t)}\mathbf{N}^{1/2}$ shows that

$$\tilde{\mathbf{D}}^{(t)-1/2}\mathbf{P}^{(t)}\tilde{\mathbf{D}}^{(t)-1/2}\mathcal{Z} = \mathcal{Z}\left(\mathbf{I} - \mathbf{N}^{-1/2}\tilde{\mathbf{M}}_{n1}^{(t)}\mathbf{N}^{1/2}\right).$$

Therefore,

$$\tilde{\mathbf{L}}_{n1}^{(t)}\tilde{\mathbf{u}}^{(i)} = \mathcal{Z}\mathbf{N}^{-1/2}\tilde{\mathbf{M}}_{n1}^{(t)}\tilde{\mathbf{x}}^{(i)}.$$

Since $\tilde{\mathbf{L}}_{n1}^{(t)}$ is symmetric,

$$\tilde{\mathbf{L}}_{n1}^{(t)\top}\tilde{\mathbf{L}}_{n1}^{(t)}\tilde{\mathbf{u}}^{(i)} = \mathcal{Z}\mathbf{N}^{-1/2}\tilde{\mathbf{M}}_{n1}^{(t)2}\tilde{\mathbf{x}}^{(i)}.$$

Summing over $t$ gives

$$\tilde{\mathcal{L}}_{n1}\tilde{\mathcal{L}}_{n1}^{\top}\tilde{\mathbf{u}}^{(i)} = \mathcal{Z}\mathbf{N}^{-1/2}\left(\sum_{t=1}^{T}\tilde{\mathbf{M}}_{n1}^{(t)2}\right)\tilde{\mathbf{x}}^{(i)}.$$

Recall $\tilde{\mathcal{M}}_{n1}$ satisfy $\tilde{\mathcal{M}}_{n1}\tilde{\mathcal{M}}_{n1}^{\top} = \sum_{t=1}^{T}\tilde{\mathbf{M}}_{n1}^{(t)2}$, and $\tilde{\mathbf{x}}^{(i)}$ is a left singular vector of $\tilde{\mathcal{M}}_{n1}$ with singular value $\tilde{\lambda}_i$. Therefore

$$\tilde{\mathcal{L}}_{n1}\tilde{\mathcal{L}}_{n1}^{\top}\tilde{\mathbf{u}}^{(i)} = \tilde{\lambda}_i^2\,\tilde{\mathbf{u}}^{(i)}.$$

Thus $\tilde{\mathbf{u}}^{(i)}$ is a left singular vector of $\tilde{\mathcal{L}}_{n1}$ with singular value $\tilde{\lambda}_i$. $\qquad\square$

**Corollary E.4** (Bounds for the population singular values $\tilde{\sigma}_i$).

$$\tilde{\sigma}_n = \mathcal{O}(1), \qquad \tilde{\sigma}_2 - \tilde{\sigma}_1 = \Omega(1), \qquad \tilde{\sigma}_{K+1} - \tilde{\sigma}_K = \Omega(1) \quad a.s.$$

*Proof.* Since the labels $\{\mathbf{z}_i\}_{i=1}^{n}$ are drawn independently with $\Pr(\mathbf{z}_i = k) = \pi_k$, the random variables $\mathbf{1}\{\mathbf{z}_i = k\}$ are i.i.d. Bernoulli($\pi_k$). Hence

$$\mathbb{E}[n_k] = \sum_{i=1}^{n}\mathbb{E}[\mathbf{1}\{\mathbf{z}_i = k\}] = \sum_{i=1}^{n}\pi_k = n\pi_k.$$

By Hoeffding's inequality, for any $s > 0$,

$$\Pr\left(\left|n_k - n\pi_k\right| \geq s\right) \leq 2\exp\left(-\frac{s^2}{n}\right).$$

Fix $\alpha > 0$ and choose $s = \sqrt{(\alpha+1)n\log n}$. Then

$$\Pr\left(\left|n_k - n\pi_k\right| \geq \sqrt{(\alpha+1)n\log n}\right) \leq 2n^{-(\alpha+1)}.$$

Applying a union bound over $k \in \{1, \ldots, K\}$ yields

$$\Pr\left(\max_{k\in\{1,\ldots,K\}}\left|n_k - n\pi_k\right| \geq \sqrt{(\alpha+1)n\log n}\right) \leq 2K\,n^{-(\alpha+1)}.$$

Since $\sum_{n=1}^{\infty} 2K\, n^{-(\alpha+1)} < \infty$, the Borel–Cantelli lemma implies

$$\max_{k \in \{1,\ldots,K\}} \left| n_k - n\pi_k \right| = \mathcal{O}\left(n^{1/2} \log^{1/2} n\right) \quad \text{a.s.}$$

Because $\pi_k > 0$ for all $k \in \{1, \ldots, K\}$, it follows that

$$n_k = n\pi_k + \mathcal{O}\left(n^{1/2} \log^{1/2} n\right) = \Omega(n) \quad \text{a.s.}$$

Consequently, the matrix $\mathbf{N} = \mathrm{diag}(n_1, \ldots, n_K)$ is invertible almost surely.

By Lemma E.3, the noise-free unfolded normalized Laplacian $\tilde{\mathcal{L}}_{\mathrm{n1}}$ has singular value $\sqrt{T}$ with multiplicity $n - K$, and its remaining $K$ singular values coincide with those of the community level matrix $\tilde{\mathcal{M}}_{\mathrm{n1}}$.

By the triangle inequality,

$$\|\tilde{\mathcal{M}}_{\mathrm{n1}} - \bar{\mathcal{M}}_{\mathrm{n1}}\|_2 \leq \sum_{t=1}^{T} \|\tilde{\mathbf{M}}_{\mathrm{n1}}^{(t)} - \bar{\mathbf{M}}_{\mathrm{n1}}^{(t)}\|_2.$$

Following the argument of Theorem 3.1 in Deng et al. (2021), for each $t \in \{1, \ldots, T\}$,

$$\|\tilde{\mathbf{M}}_{\mathrm{n1}}^{(t)} - \bar{\mathbf{M}}_{\mathrm{n1}}^{(t)}\|_2 \leq \frac{\|\tilde{\mathbf{Q}}^{(t)} - \bar{\mathbf{Q}}^{(t)}\|_2}{\tilde{d}^{(t)}}$$
$$+ \frac{K\left(\|\tilde{\mathbf{Q}}^{(t)} - \bar{\mathbf{Q}}^{(t)}\|_2 + 2\|\bar{\mathbf{Q}}^{(t)}\|_2\right)\|\tilde{\mathbf{Q}}^{(t)} - \bar{\mathbf{Q}}^{(t)}\|_2}{2\min(\tilde{d}^{(t)}, \bar{d}^{(t)})^3},$$

where $\tilde{d}^{(t)}$ and $\bar{d}^{(t)}$ denote the minimum degrees of $\tilde{\mathbf{Q}}^{(t)}$ and $\bar{\mathbf{Q}}^{(t)}$, respectively.

Since $\bar{\mathbf{Q}}^{(t)}$ depends only on $\boldsymbol{\pi}$ and $\mathbf{B}^{(t)}$, we have

$$\|\bar{\mathbf{Q}}^{(t)}\|_2 = \mathcal{O}(1), \qquad \bar{d}^{(t)} = \Omega(1).$$

Moreover, using $n_k = \Omega(n)$ a.s. and $\mathbf{B}_{\min}^{(t)} > 0$, we obtain

$$\tilde{d}^{(t)} = \min_{k \in \{1,\ldots,K\}} \sum_{\ell=1}^{K} \frac{n_k}{n} \frac{n_\ell}{n} \mathbf{B}_{c^{(t)}(k)c^{(t)}(\ell)}^{(t)}$$
$$\geq \mathbf{B}_{\min}^{(t)} \frac{\min_{k \in \{1,\ldots,K\}} n_k}{n} = \Omega(1) \quad \text{a.s.}$$

Next, since $\|\cdot\|_2 \leq \|\cdot\|_F$,

$$\|\tilde{\mathbf{Q}}^{(t)} - \bar{\mathbf{Q}}^{(t)}\|_2 \leq \left(\sum_{k=1}^{K} \sum_{\ell=1}^{K} \left(\frac{n_k}{n} \frac{n_\ell}{n} \mathbf{B}_{c^{(t)}(k)c^{(t)}(\ell)}^{(t)} - \pi_k \pi_\ell \mathbf{B}_{c^{(t)}(k)c^{(t)}(\ell)}^{(t)}\right)^2\right)^{1/2}$$
$$= 2K \frac{\max_{k \in \{1,\ldots,K\}} |n_k - n\pi_k|}{n}.$$

Using the earlier concentration bound,

$$\|\tilde{\mathbf{Q}}^{(t)} - \bar{\mathbf{Q}}^{(t)}\|_2 = \mathcal{O}\left(\frac{\log^{1/2} n}{n^{1/2}}\right) \quad \text{a.s.}$$

Substituting this into the previous inequality yields

$$\|\tilde{\mathbf{M}}_{\mathrm{n1}}^{(t)} - \bar{\mathbf{M}}_{\mathrm{n1}}^{(t)}\|_2 = \mathcal{O}\left(\frac{\log^{1/2} n}{n^{1/2}}\right) \quad \text{a.s.}$$

and hence, since $T$ is fixed,

$$\|\tilde{\mathcal{M}}_{\mathrm{n}1} - \bar{\mathcal{M}}_{\mathrm{n}1}\|_2 = \mathcal{O}\left(\frac{\log^{1/2} n}{n^{1/2}}\right) \quad \text{a.s.}$$

By Weyl's inequality,

$$|\tilde{\lambda}_i - \bar{\lambda}_i| \le \|\tilde{\mathcal{M}}_{\mathrm{n}1} - \bar{\mathcal{M}}_{\mathrm{n}1}\|_2, \qquad i \in \{1, \dots, K\}.$$

Since $0 \le \bar{\lambda}_1 < \bar{\lambda}_2 \le \cdots \le \bar{\lambda}_K < \sqrt{T}$, define

$$\Delta_{12} := \bar{\lambda}_2 - \bar{\lambda}_1 > 0, \qquad \Delta_{K,T} := \sqrt{T} - \bar{\lambda}_K > 0.$$

On the almost sure event where $\|\tilde{\mathcal{M}}_{\mathrm{n}1} - \bar{\mathcal{M}}_{\mathrm{n}1}\|_2 \to 0$, there exists $n_0$ such that for all $n \ge n_0$,

$$\|\tilde{\mathcal{M}}_{\mathrm{n}1} - \bar{\mathcal{M}}_{\mathrm{n}1}\|_2 \le \min\left\{\frac{\Delta_{12}}{3}, \frac{\Delta_{K,T}}{2}\right\}.$$

Hence, for all $n \ge n_0$,

$$\tilde{\sigma}_1 < \frac{2\bar{\lambda}_1 + \bar{\lambda}_2}{3}, \qquad \frac{\bar{\lambda}_1 + 2\bar{\lambda}_2}{3} < \tilde{\sigma}_i < \frac{\bar{\lambda}_K + \sqrt{T}}{2}, \quad i = 2, \dots, K,$$

and $\tilde{\sigma}_{K+1} = \cdots = \tilde{\sigma}_n = \sqrt{T}$. This completes the proof. $\qquad\square$

**Corollary E.5** (Bounds for the empirical singular values $\sigma_i$).

$$\sigma_n = \mathcal{O}(1), \qquad \sigma_2 = \Omega(1) \quad a.s.$$

*Proof.* By Weyl's inequality,

$$\tilde{\sigma}_i - \|\mathcal{L}_{\mathrm{n}1} - \tilde{\mathcal{L}}_{\mathrm{n}1}\|_2 \le \sigma_i \le \tilde{\sigma}_i + \|\mathcal{L}_{\mathrm{n}1} - \tilde{\mathcal{L}}_{\mathrm{n}1}\|_2.$$

Applying Lemma E.2 and Corollary E.4 yields the claim. $\qquad\square$

**Lemma E.6** (Deviation bound for the projection matrices). *Let $\mathbf{U}, \mathbf{V} \in \mathbb{R}^{n \times (K-1)}$ denote the empirical left and right singular vector matrices associated with the informative singular values of the unfolded operator, and let $\tilde{\mathbf{U}}, \tilde{\mathbf{V}}$ denote their population counterparts. Then*

$$\left\|\mathbf{U}\mathbf{U}^\top - \tilde{\mathbf{U}}\tilde{\mathbf{U}}^\top\right\|_2 = \mathcal{O}\left(\frac{1}{\rho^{1/2}n^{1/2}}\right) \quad a.s.,$$

$$\left\|\mathbf{V}\mathbf{V}^\top - \tilde{\mathbf{V}}\tilde{\mathbf{V}}^\top\right\|_2 = \mathcal{O}\left(\frac{1}{\rho^{1/2}n^{1/2}}\right) \quad a.s.$$

*Proof.* The unfolded normalized Laplacian $\mathcal{L}$ and its population counterpart $\tilde{\mathcal{L}}_{\mathrm{n}1}$ satisfy the deviation bound $\|\mathcal{L}_{\mathrm{n}1} - \tilde{\mathcal{L}}_{\mathrm{n}1}\|_2 = \mathcal{O}((\rho n)^{-1/2})$ almost surely by Lemma E.2. Moreover, Corollary E.4 ensures that the informative singular values are separated from the remainder of the spectrum by a constant gap. Applying Theorem 4 of Yu et al. (2015) to the singular subspaces corresponding to the $(K-1)$ informative components yields

$$\left\|\mathbf{U}\mathbf{U}^\top - \tilde{\mathbf{U}}\tilde{\mathbf{U}}^\top\right\|_2 \le \frac{2\sqrt{K-1}\left(2\sigma_n + \|\mathcal{L}_{\mathrm{n}1} - \tilde{\mathcal{L}}_{\mathrm{n}1}\|_2\right)\|\mathcal{L}_{\mathrm{n}1} - \tilde{\mathcal{L}}_{\mathrm{n}1}\|_2}{\min\left(\sigma_{K+1}^2 - \sigma_K^2, \; \sigma_2^2 - \sigma_1^2\right)}.$$

The denominator is bounded below by a positive constant due to spectral separation, while $\sigma_n = \mathcal{O}(1)$ and $\|\mathcal{L}_{\mathrm{n}1} - \tilde{\mathcal{L}}_{\mathrm{n}1}\|_2 = \mathcal{O}((\rho n)^{-1/2})$. Substituting these bounds gives

$$\left\|\mathbf{U}\mathbf{U}^\top - \tilde{\mathbf{U}}\tilde{\mathbf{U}}^\top\right\|_2 = \mathcal{O}\left(\frac{1}{\rho^{1/2}n^{1/2}}\right) \quad \text{a.s.}$$

The same argument applies verbatim to $\left\|\mathbf{V}\mathbf{V}^\top - \tilde{\mathbf{V}}\tilde{\mathbf{V}}^\top\right\|_2$. $\qquad\square$

**Lemma E.7** (Existence of a common orthogonal alignment)**.** *There exists an orthogonal matrix* $\mathbf{W} \in \mathbb{O}\left((K-1) \times (K-1)\right)$ *such that*

$$\left\|\tilde{\mathbf{U}}^\top \mathbf{U} - \mathbf{W}\right\|_F = \mathcal{O}\left(\frac{1}{\rho^{1/2} n^{1/2}}\right) \quad a.s.,$$

$$\left\|\tilde{\mathbf{V}}^\top \mathbf{V} - \mathbf{W}\right\|_F = \mathcal{O}\left(\frac{1}{\rho^{1/2} n^{1/2}}\right) \quad a.s.$$

*Proof.* Consider the matrix $\tilde{\mathbf{U}}^\top \mathbf{U} + \tilde{\mathbf{V}}^\top \mathbf{V}$ and let its singular value decomposition be

$$\tilde{\mathbf{U}}^\top \mathbf{U} + \tilde{\mathbf{V}}^\top \mathbf{V} = \mathbf{W}_1 \mathbf{\Sigma}' \mathbf{W}_2^\top,$$

where $\mathbf{W}_1, \mathbf{W}_2 \in \mathbb{O}\left((K-1) \times (K-1)\right)$. Define $\mathbf{W} = \mathbf{W}_1 \mathbf{W}_2^\top$, which belongs to $\mathbb{O}\left((K-1) \times (K-1)\right)$. This choice minimizes

$$\left\|\tilde{\mathbf{U}}^\top \mathbf{U} - \mathbf{Q}\right\|_F + \left\|\tilde{\mathbf{V}}^\top \mathbf{V} - \mathbf{Q}\right\|_F$$

over all $\mathbf{Q} \in \mathbb{O}\left((K-1) \times (K-1)\right)$.

Next, write the singular value decomposition of $\tilde{\mathbf{U}}^\top \mathbf{U}$ as

$$\tilde{\mathbf{U}}^\top \mathbf{U} = \mathbf{W}_{\mathbf{U},1} \mathbf{\Sigma}'_{\mathbf{U}} \mathbf{W}_{\mathbf{U},2}^\top,$$

and define

$$\mathbf{W}_{\mathbf{U}} = \mathbf{W}_{\mathbf{U},1} \mathbf{W}_{\mathbf{U},2}^\top \in \mathbb{O}\left((K-1) \times (K-1)\right).$$

Let the singular values be $\sigma'_k = \cos(\theta_k)$. Then

$$\left\|\tilde{\mathbf{U}}^\top \mathbf{U} - \mathbf{W}_{\mathbf{U}}\right\|_F = \left\|\mathbf{\Sigma}'_{\mathbf{U}} - \mathbf{I}\right\|_F$$

$$= \sqrt{\sum_{k=1}^{K-1} \left(1 - \sigma'_k\right)^2} \leq \sum_{k=1}^{K-1} \left(1 - \sigma'_k\right) \leq \sum_{k=1}^{K-1} \left(1 - \sigma'^2_k\right)$$

$$= \sum_{k=1}^{K-1} \sin^2 \theta_k \leq (K-1) \left\|\mathbf{U}\mathbf{U}^\top - \tilde{\mathbf{U}}\tilde{\mathbf{U}}^\top\right\|_2^2.$$

Similarly,

$$\left\|\tilde{\mathbf{V}}^\top \mathbf{V} - \mathbf{W}_{\mathbf{U}}\right\|_F \leq \left\|\tilde{\mathbf{V}}^\top \mathbf{V} - \tilde{\mathbf{U}}^\top \mathbf{U}\right\|_F + \left\|\tilde{\mathbf{U}}^\top \mathbf{U} - \mathbf{W}_{\mathbf{U}}\right\|_F$$

$$\leq (K-1) \left\|\tilde{\mathbf{V}}^\top \mathbf{V} - \tilde{\mathbf{U}}^\top \mathbf{U}\right\|_2 + (K-1) \left\|\mathbf{U}\mathbf{U}^\top - \tilde{\mathbf{U}}\tilde{\mathbf{U}}^\top\right\|_2^2.$$

Therefore,

$$\left\|\tilde{\mathbf{U}}^\top \mathbf{U} - \mathbf{W}\right\|_F + \left\|\tilde{\mathbf{V}}^\top \mathbf{V} - \mathbf{W}\right\|_F \leq (K-1) \left\|\tilde{\mathbf{V}}^\top \mathbf{V} - \tilde{\mathbf{U}}^\top \mathbf{U}\right\|_2 + 2(K-1) \left\|\mathbf{U}\mathbf{U}^\top - \tilde{\mathbf{U}}\tilde{\mathbf{U}}^\top\right\|_2^2.$$

By Lemma E.6, both terms on the right-hand side are of order $(\rho n)^{-1/2}$ almost surely, which completes the proof. $\quad\square$

**Corollary E.8.**

$$\|\tilde{\mathbf{U}}^\top \mathbf{U} \mathbf{\Sigma} - \tilde{\mathbf{\Sigma}} \tilde{\mathbf{V}}^\top \mathbf{V}\|_F = \mathcal{O}\left(\frac{1}{\rho^{1/2} n^{1/2}}\right), \qquad \|\tilde{\mathbf{V}}^\top \mathbf{V} \mathbf{\Sigma} - \tilde{\mathbf{\Sigma}} \tilde{\mathbf{U}}^\top \mathbf{U}\|_F = \mathcal{O}\left(\frac{1}{\rho^{1/2} n^{1/2}}\right) \quad a.s.$$

*Moreover,*

$$\|\tilde{\mathbf{U}}^\top \mathbf{U} - \tilde{\mathbf{V}}^\top \mathbf{V}\|_F = \mathcal{O}\left(\frac{1}{\rho^{1/2} n^{1/2}}\right) \quad a.s.$$

*Proof.* By Lemma E.2, we have

$$
\begin{aligned}
\|\tilde{\mathbf{U}}^\top \mathbf{U}\boldsymbol{\Sigma} - \tilde{\boldsymbol{\Sigma}}\tilde{\mathbf{V}}^\top \mathbf{V}\|_F &= \|\tilde{\mathbf{U}}^\top(\mathcal{L}_{\mathrm{n1}} - \tilde{\mathcal{L}}_{\mathrm{n1}})\mathbf{V}\|_F \\
&\leq \sqrt{K-1}\,\|\tilde{\mathbf{U}}^\top(\mathcal{L}_{\mathrm{n1}} - \tilde{\mathcal{L}}_{\mathrm{n1}})\mathbf{V}\|_2 \\
&\leq \sqrt{K-1}\,\|\mathcal{L}_{\mathrm{n1}} - \tilde{\mathcal{L}}_{\mathrm{n1}}\|_2 \\
&= \mathcal{O}\left(\frac{1}{\rho^{1/2}n^{1/2}}\right) \quad \text{a.s.}
\end{aligned}
$$

The same argument applies to $\|\tilde{\boldsymbol{\Sigma}}\tilde{\mathbf{U}}^\top \mathbf{U} - \tilde{\mathbf{V}}^\top \mathbf{V}\boldsymbol{\Sigma}\|_F$.

Next, observe that

$$
\tilde{\mathbf{U}}^\top \mathbf{U} - \tilde{\mathbf{V}}^\top \mathbf{V} = \left[(\tilde{\mathbf{U}}^\top \mathbf{U}\boldsymbol{\Sigma} - \tilde{\boldsymbol{\Sigma}}\tilde{\mathbf{V}}^\top \mathbf{V}) + (\tilde{\boldsymbol{\Sigma}}\tilde{\mathbf{U}}^\top \mathbf{U} - \tilde{\mathbf{V}}^\top \mathbf{V}\boldsymbol{\Sigma})\right]\boldsymbol{\Sigma}^{-1} - \tilde{\boldsymbol{\Sigma}}(\tilde{\mathbf{U}}^\top \mathbf{U} - \tilde{\mathbf{V}}^\top \mathbf{V})\boldsymbol{\Sigma}^{-1}.
$$

Therefore, for each pair of indices $(i,j)$,

$$
\begin{aligned}
\left|(\tilde{\mathbf{U}}^\top \mathbf{U} - \tilde{\mathbf{V}}^\top \mathbf{V})_{ij}\right| &\leq \left|(\tilde{\mathbf{U}}^\top \mathbf{U} - \tilde{\mathbf{V}}^\top \mathbf{V})_{ij}\right|\left(1 + \frac{\tilde{\sigma}_i}{\sigma_j}\right) \\
&\leq \left(\|\tilde{\mathbf{U}}^\top \mathbf{U}\boldsymbol{\Sigma} - \tilde{\boldsymbol{\Sigma}}\tilde{\mathbf{V}}^\top \mathbf{V}\|_F + \|\tilde{\boldsymbol{\Sigma}}\tilde{\mathbf{U}}^\top \mathbf{U} - \tilde{\mathbf{V}}^\top \mathbf{V}\boldsymbol{\Sigma}\|_F\right)\|\boldsymbol{\Sigma}^{-1}\|_2.
\end{aligned}
$$

Taking the Frobenius norm on both sides yields

$$
\|\tilde{\mathbf{U}}^\top \mathbf{U} - \tilde{\mathbf{V}}^\top \mathbf{V}\|_F \leq \left(\|\tilde{\mathbf{U}}^\top \mathbf{U}\boldsymbol{\Sigma} - \tilde{\boldsymbol{\Sigma}}\tilde{\mathbf{V}}^\top \mathbf{V}\|_F + \|\tilde{\boldsymbol{\Sigma}}\tilde{\mathbf{U}}^\top \mathbf{U} - \tilde{\mathbf{V}}^\top \mathbf{V}\boldsymbol{\Sigma}\|_F\right)\|\boldsymbol{\Sigma}^{-1}\|_2.
$$

From the first part of the proof and Corollary E.5, we have

$$
\|\boldsymbol{\Sigma}^{-1}\|_2 = \mathcal{O}(1) \quad \text{a.s.},
$$

and therefore

$$
\|\tilde{\mathbf{U}}^\top \mathbf{U} - \tilde{\mathbf{V}}^\top \mathbf{V}\|_F = \mathcal{O}\left(\frac{1}{\rho^{1/2}n^{1/2}}\right) \quad \text{a.s.}
$$

This completes the proof. □

**Corollary E.9.**

$$
\|\mathbf{U} - \tilde{\mathbf{U}}\tilde{\mathbf{U}}^\top \mathbf{U}\|_2, \qquad \|\mathbf{V} - \tilde{\mathbf{V}}\tilde{\mathbf{V}}^\top \mathbf{V}\|_2 = \mathcal{O}\left(\frac{1}{\rho^{1/2}n^{1/2}}\right) \quad a.s.
$$

*Proof.* By Lemma E.6,

$$
\begin{aligned}
\|\mathbf{U} - \tilde{\mathbf{U}}\tilde{\mathbf{U}}^\top \mathbf{U}\|_2 &= \|(\mathbf{U}\mathbf{U}^\top - \tilde{\mathbf{U}}\tilde{\mathbf{U}}^\top)\mathbf{U}\|_2 \\
&\leq \|\mathbf{U}\mathbf{U}^\top - \tilde{\mathbf{U}}\tilde{\mathbf{U}}^\top\|_2\,\|\mathbf{U}\|_2 \\
&\leq \|\mathbf{U}\mathbf{U}^\top - \tilde{\mathbf{U}}\tilde{\mathbf{U}}^\top\|_2 \\
&\leq \|\mathbf{U}\mathbf{U}^\top - \tilde{\mathbf{U}}\tilde{\mathbf{U}}^\top\|_F \\
&= \mathcal{O}\left(\frac{1}{\rho^{1/2}n^{1/2}}\right) \quad \text{a.s.}
\end{aligned}
$$

where we used $\|\mathbf{U}\|_2 = 1$. The same argument applies to $\mathbf{V}$, since $\|\mathbf{V}\|_2 = 1$. □

**Corollary E.10.**

$$
\|\mathbf{U} - \tilde{\mathbf{U}}\mathbf{W}\|_2, \qquad \|\mathbf{V} - \tilde{\mathbf{V}}\mathbf{W}\|_2 = \mathcal{O}\left(\frac{1}{\rho^{1/2}n^{1/2}}\right) \quad a.s.
$$

*Proof.* Write
$$\mathbf{U} - \tilde{\mathbf{U}}\mathbf{W} = (\mathbf{U} - \tilde{\mathbf{U}}\tilde{\mathbf{U}}^\top\mathbf{U}) + \tilde{\mathbf{U}}(\tilde{\mathbf{U}}^\top\mathbf{U} - \mathbf{W}).$$

Taking spectral norms and applying the triangle inequality gives
$$\|\mathbf{U} - \tilde{\mathbf{U}}\mathbf{W}\|_2 \le \|\mathbf{U} - \tilde{\mathbf{U}}\tilde{\mathbf{U}}^\top\mathbf{U}\|_2 + \|\tilde{\mathbf{U}}\|_2\,\|\tilde{\mathbf{U}}^\top\mathbf{U} - \mathbf{W}\|_2.$$

Since $\tilde{\mathbf{U}}$ has orthonormal columns, $\|\tilde{\mathbf{U}}\|_2 = 1$, and $\|\tilde{\mathbf{U}}^\top\mathbf{U} - \mathbf{W}\|_2 \le \|\tilde{\mathbf{U}}^\top\mathbf{U} - \mathbf{W}\|_F$. Therefore,
$$\|\mathbf{U} - \tilde{\mathbf{U}}\mathbf{W}\|_2 \le \|\mathbf{U} - \tilde{\mathbf{U}}\tilde{\mathbf{U}}^\top\mathbf{U}\|_2 + \|\tilde{\mathbf{U}}^\top\mathbf{U} - \mathbf{W}\|_F.$$

The first term is $\mathcal{O}((\rho n)^{-1/2})$ a.s. by Corollary E.9, and the second term is $\mathcal{O}((\rho n)^{-1/2})$ a.s. by Lemma E.7, proving the claim. The same argument applies to $\mathbf{V}$. $\qquad\square$

**Corollary E.11.** *Almost surely, the following bounds hold:*

1. $\|\mathbf{W}\boldsymbol{\Sigma} - \tilde{\boldsymbol{\Sigma}}\mathbf{W}\|_F = \mathcal{O}((\rho n)^{-1/2})$,

2. $\|\mathbf{W}\boldsymbol{\Sigma}^{1/2} - \tilde{\boldsymbol{\Sigma}}^{1/2}\mathbf{W}\|_F = \mathcal{O}((\rho n)^{-1/2})$,

3. $\|\mathbf{W}\boldsymbol{\Sigma}^{-1/2} - \tilde{\boldsymbol{\Sigma}}^{-1/2}\mathbf{W}\|_F = \mathcal{O}((\rho n)^{-1/2})$.

*Proof.* 1. Decompose
$$\mathbf{W}\boldsymbol{\Sigma} - \tilde{\boldsymbol{\Sigma}}\mathbf{W} = (\mathbf{W} - \tilde{\mathbf{U}}^\top\mathbf{U})\boldsymbol{\Sigma} + (\tilde{\mathbf{U}}^\top\mathbf{U}\boldsymbol{\Sigma} - \tilde{\boldsymbol{\Sigma}}\tilde{\mathbf{V}}^\top\mathbf{V}) + \tilde{\boldsymbol{\Sigma}}(\tilde{\mathbf{V}}^\top\mathbf{V} - \mathbf{W}).$$

By the triangle inequality and $\|AB\|_F \le \|A\|_F\|B\|_2$,
$$\|\mathbf{W}\boldsymbol{\Sigma} - \tilde{\boldsymbol{\Sigma}}\mathbf{W}\|_F \le \|\mathbf{W} - \tilde{\mathbf{U}}^\top\mathbf{U}\|_F\,\|\boldsymbol{\Sigma}\|_2 + \|\tilde{\mathbf{U}}^\top\mathbf{U}\boldsymbol{\Sigma} - \tilde{\boldsymbol{\Sigma}}\tilde{\mathbf{V}}^\top\mathbf{V}\|_F$$
$$+ \|\tilde{\boldsymbol{\Sigma}}\|_2\,\|\tilde{\mathbf{V}}^\top\mathbf{V} - \mathbf{W}\|_F.$$

Using $\|\boldsymbol{\Sigma}\|_2 = \sigma_n = \mathcal{O}(1)$ and $\|\tilde{\boldsymbol{\Sigma}}\|_2 = \tilde{\sigma}_n = \mathcal{O}(1)$ (Corollaries E.4 and E.5), together with Lemma E.7 and Corollary E.8, yields
$$\|\mathbf{W}\boldsymbol{\Sigma} - \tilde{\boldsymbol{\Sigma}}\mathbf{W}\|_F = \mathcal{O}((\rho n)^{-1/2}) \quad \text{a.s.}$$

2. Let $\sigma_{\min} := \sigma_2$ and $\tilde{\sigma}_{\min} := \tilde{\sigma}_2$. By Corollary E.5, $\sigma_{\min} = \Omega(1)$ a.s., and by Corollary E.4, $\tilde{\sigma}_{\min} = \Omega(1)$ a.s. For each $i, j$,
$$\big|(\mathbf{W}\boldsymbol{\Sigma}^{1/2} - \tilde{\boldsymbol{\Sigma}}^{1/2}\mathbf{W})_{ij}\big| = |W_{ij}|\,|\sigma_j^{1/2} - \tilde{\sigma}_i^{1/2}|$$
$$= |W_{ij}|\,\frac{|\sigma_j - \tilde{\sigma}_i|}{\sigma_j^{1/2} + \tilde{\sigma}_i^{1/2}}$$
$$\le \frac{|(\mathbf{W}\boldsymbol{\Sigma} - \tilde{\boldsymbol{\Sigma}}\mathbf{W})_{ij}|}{\sigma_{\min}^{1/2} + \tilde{\sigma}_{\min}^{1/2}}.$$

Squaring and summing over $i, j$ gives
$$\|\mathbf{W}\boldsymbol{\Sigma}^{1/2} - \tilde{\boldsymbol{\Sigma}}^{1/2}\mathbf{W}\|_F \le \frac{\|\mathbf{W}\boldsymbol{\Sigma} - \tilde{\boldsymbol{\Sigma}}\mathbf{W}\|_F}{\sigma_{\min}^{1/2} + \tilde{\sigma}_{\min}^{1/2}} = \mathcal{O}((\rho n)^{-1/2}) \quad \text{a.s.,}$$

using part (1) and $\sigma_{\min}^{1/2} + \tilde{\sigma}_{\min}^{1/2} = \Omega(1)$ a.s.

3. Similarly, for each $i, j$,
$$\big|(\mathbf{W}\boldsymbol{\Sigma}^{-1/2} - \tilde{\boldsymbol{\Sigma}}^{-1/2}\mathbf{W})_{ij}\big| = |W_{ij}|\,|\sigma_j^{-1/2} - \tilde{\sigma}_i^{-1/2}|$$
$$= |W_{ij}|\,\frac{|\sigma_j^{1/2} - \tilde{\sigma}_i^{1/2}|}{\sigma_j^{1/2}\tilde{\sigma}_i^{1/2}}$$
$$\le \frac{|(\mathbf{W}\boldsymbol{\Sigma}^{1/2} - \tilde{\boldsymbol{\Sigma}}^{1/2}\mathbf{W})_{ij}|}{\sigma_{\min}^{1/2}\tilde{\sigma}_{\min}^{1/2}}.$$

Therefore,

$$\|\mathbf{W}\boldsymbol{\Sigma}^{-1/2} - \tilde{\boldsymbol{\Sigma}}^{-1/2}\mathbf{W}\|_F \leq \frac{\|\mathbf{W}\boldsymbol{\Sigma}^{1/2} - \tilde{\boldsymbol{\Sigma}}^{1/2}\mathbf{W}\|_F}{\sigma_{\min}^{1/2}\tilde{\sigma}_{\min}^{1/2}} = \mathcal{O}((\rho n)^{-1/2}) \quad \text{a.s.,}$$

since $\sigma_{\min}^{1/2}\tilde{\sigma}_{\min}^{1/2} = \Omega(1)$ a.s. and part (2) holds.

$\square$

Finally, we complete the proof of Theorem 2.

*Proof.* By Corollary E.5, Corollary E.10, and Corollary E.11, we bound

$$\|\hat{\mathbf{Y}}^{(t)} - \tilde{\mathbf{Y}}^{(t)}\mathbf{W}\|_2 = \left\|(\mathbf{V}^{(t)}\boldsymbol{\Sigma}^{1/2} - \mathbf{U}\boldsymbol{\Sigma}^{-1/2}) - (\tilde{\mathbf{V}}^{(t)}\tilde{\boldsymbol{\Sigma}}^{1/2} - \tilde{\mathbf{U}}\tilde{\boldsymbol{\Sigma}}^{-1/2})\mathbf{W}\right\|_2$$
$$\leq \|\mathbf{V}^{(t)}\boldsymbol{\Sigma}^{1/2} - \tilde{\mathbf{V}}^{(t)}\tilde{\boldsymbol{\Sigma}}^{1/2}\mathbf{W}\|_2 + \|\mathbf{U}\boldsymbol{\Sigma}^{-1/2} - \tilde{\mathbf{U}}\tilde{\boldsymbol{\Sigma}}^{-1/2}\mathbf{W}\|_2.$$

We control these two terms separately.

**Step 1: Bounding the $\mathbf{V}^{(t)}$ term.** Using the fact that $\mathbf{V}^{(t)}$ is a block of $\mathbf{V}$,

$$\|\mathbf{V}^{(t)}\boldsymbol{\Sigma}^{1/2} - \tilde{\mathbf{V}}^{(t)}\tilde{\boldsymbol{\Sigma}}^{1/2}\mathbf{W}\|_2 \leq \|\mathbf{V}\boldsymbol{\Sigma}^{1/2} - \tilde{\mathbf{V}}\tilde{\boldsymbol{\Sigma}}^{1/2}\mathbf{W}\|_2$$
$$= \|(\mathbf{V} - \tilde{\mathbf{V}}\mathbf{W})\boldsymbol{\Sigma}^{1/2} + \tilde{\mathbf{V}}(\mathbf{W}\boldsymbol{\Sigma}^{1/2} - \tilde{\boldsymbol{\Sigma}}^{1/2}\mathbf{W})\|_2$$
$$\leq \|\mathbf{V} - \tilde{\mathbf{V}}\mathbf{W}\|_2\,\|\boldsymbol{\Sigma}^{1/2}\|_2 + \|\tilde{\mathbf{V}}\|_2\,\|\mathbf{W}\boldsymbol{\Sigma}^{1/2} - \tilde{\boldsymbol{\Sigma}}^{1/2}\mathbf{W}\|_2$$
$$\leq \sigma_n^{1/2}\|\mathbf{V} - \tilde{\mathbf{V}}\mathbf{W}\|_2 + \|\mathbf{W}\boldsymbol{\Sigma}^{1/2} - \tilde{\boldsymbol{\Sigma}}^{1/2}\mathbf{W}\|_F$$
$$= \mathcal{O}\left((\rho n)^{-1/2}\right) \quad \text{a.s.,}$$

where we used $\|\tilde{\mathbf{V}}\|_2 = 1$ and $\|\cdot\|_2 \leq \|\cdot\|_F$.

**Step 2: Bounding the U term.** Similarly,

$$\|\mathbf{U}\boldsymbol{\Sigma}^{-1/2} - \tilde{\mathbf{U}}\tilde{\boldsymbol{\Sigma}}^{-1/2}\mathbf{W}\|_2 = \|(\mathbf{U} - \tilde{\mathbf{U}}\mathbf{W})\boldsymbol{\Sigma}^{-1/2} + \tilde{\mathbf{U}}(\mathbf{W}\boldsymbol{\Sigma}^{-1/2} - \tilde{\boldsymbol{\Sigma}}^{-1/2}\mathbf{W})\|_2$$
$$\leq \|\mathbf{U} - \tilde{\mathbf{U}}\mathbf{W}\|_2\,\|\boldsymbol{\Sigma}^{-1/2}\|_2 + \|\tilde{\mathbf{U}}\|_2\,\|\mathbf{W}\boldsymbol{\Sigma}^{-1/2} - \tilde{\boldsymbol{\Sigma}}^{-1/2}\mathbf{W}\|_2$$
$$\leq \sigma_n^{-1/2}\|\mathbf{U} - \tilde{\mathbf{U}}\mathbf{W}\|_2 + \|\mathbf{W}\boldsymbol{\Sigma}^{-1/2} - \tilde{\boldsymbol{\Sigma}}^{-1/2}\mathbf{W}\|_F$$
$$= \mathcal{O}\left((\rho n)^{-1/2}\right) \quad \text{a.s.,}$$

where we used $\|\tilde{\mathbf{U}}\|_2 = 1$, $\|\cdot\|_2 \leq \|\cdot\|_F$, and Corollary E.5 to ensure $\|\boldsymbol{\Sigma}^{-1/2}\|_2 = \sigma_2^{-1/2} = \mathcal{O}(1)$ almost surely.

Combining the two bounds gives

$$\|\hat{\mathbf{Y}}^{(t)} - \tilde{\mathbf{Y}}^{(t)}\mathbf{W}\|_2 = \mathcal{O}\left((\rho n)^{-1/2}\right) \quad \text{a.s.}$$

$\square$

# F. Proof of Theorem 4.1

*Proof.* Using the identity $\tilde{\mathcal{L}}_{\text{n1}}^{\top}\tilde{\mathbf{U}} = \tilde{\mathbf{V}}\tilde{\boldsymbol{\Sigma}}$, the noise-free dynamic embedding admits the representation

$$\tilde{\mathbf{Y}}^{(t)} = \tilde{\mathbf{V}}^{(t)}\tilde{\boldsymbol{\Sigma}}^{1/2} - \tilde{\mathbf{U}}\tilde{\boldsymbol{\Sigma}}^{-1/2}$$
$$= \tilde{\mathbf{L}}_{\text{n1}}^{(t)}\tilde{\mathbf{U}}\tilde{\boldsymbol{\Sigma}}^{-1/2} - \tilde{\mathbf{U}}\tilde{\boldsymbol{\Sigma}}^{-1/2}$$
$$= (\mathbf{I} - \tilde{\mathbf{D}}^{(t)-1/2}\mathbf{P}^{(t)}\tilde{\mathbf{D}}^{(t)-1/2})\tilde{\mathbf{U}}\tilde{\boldsymbol{\Sigma}}^{-1/2} - \tilde{\mathbf{U}}\tilde{\boldsymbol{\Sigma}}^{-1/2}$$
$$= -\tilde{\mathbf{D}}^{(t)-1/2}\mathbf{P}^{(t)}\tilde{\mathbf{D}}^{(t)-1/2}\tilde{\mathbf{U}}\tilde{\boldsymbol{\Sigma}}^{-1/2}.$$

Consequently, for each $i \in \{1, \ldots, n\}$,

$$\tilde{\mathbf{Y}}_{i:}^{(t)} = -\tilde{\mathbf{d}}_i^{(t)-1/2} \mathbf{P}_{i:}^{(t)} \tilde{\mathbf{D}}^{(t)-1/2} \tilde{\mathbf{U}} \tilde{\mathbf{\Sigma}}^{-1/2}$$
$$= -(\mathbf{P}_{i:}^{(t)\top} \mathbf{1})^{-1/2} \mathbf{P}_{i:}^{(t)} \tilde{\mathbf{D}}^{(t)-1/2} \tilde{\mathbf{U}} \tilde{\mathbf{\Sigma}}^{-1/2}.$$

This expression depends on node $i$ only through its population connectivity profile $\mathbf{P}_{i:}^{(t)}$ and the population degree matrix $\tilde{\mathbf{D}}^{(t)}$. It follows immediately that

$$\mathbf{P}_{i:}^{(t)} = \mathbf{P}_{j:}^{(t)} \implies \tilde{\mathbf{Y}}_{i:}^{(t)} = \tilde{\mathbf{Y}}_{j:}^{(t)}, \qquad \mathbf{P}_{i:}^{(t)} = \mathbf{P}_{i:}^{(s)}, \ \tilde{\mathbf{D}}^{(t)} = \tilde{\mathbf{D}}^{(s)} \implies \tilde{\mathbf{Y}}_{i:}^{(t)} = \tilde{\mathbf{Y}}_{i:}^{(s)}.$$

Therefore, the noise-free dynamic embedding satisfies both cross-sectional and longitudinal stability. $\square$

When nodes with identical population connectivity profiles tend to connect more strongly to each other at a given time step (i.e., $\mathbf{P}_{i:}^{(t)} = \mathbf{P}_{j:}^{(t)}$ implies large $\mathbf{P}_{ij}^{(t)}$), mild violations of the equal-degree condition have limited impact. In this regime, the discrepancy between $\hat{\mathbf{Y}}_{i:}^{(t)}$ and $\hat{\mathbf{Y}}_{i:}^{(s)}$ is governed by the difference between $\mathbf{P}_{i:}^{(t)} \mathbf{D}^{(t)-1/2}$ and $\mathbf{P}_{i:}^{(s)} \mathbf{D}^{(s)-1/2}$. The $j$th coordinate of this difference vanishes when $\mathbf{P}_{i:}^{(t)} = \mathbf{P}_{j:}^{(t)}$ and remains small otherwise, since $\mathbf{P}_{ij}^{(t)}$ is small in that case.

## G. Proof of Proposition 4.9

*Proof.* Each matrix $\mathbf{L}_{n1}^{(t)} \mathbf{L}_{n1}^{(t)\top}$ is symmetric and positive semidefinite. By Weyl's inequality,

$$\sigma_k = \sqrt{\lambda_k \left( \sum_{t=1}^{T} \mathbf{L}_{n1}^{(t)} \mathbf{L}_{n1}^{(t)\top} \right)} \geq \sqrt{\lambda_k \left( \mathbf{L}_{n1}^{(t)} \mathbf{L}_{n1}^{(t)\top} \right) + \lambda_1 \left( \sum_{s \neq t} \mathbf{L}_{n1}^{(s)} \mathbf{L}_{n1}^{(s)\top} \right)}.$$

Since $\sum_{s \neq t} \mathbf{L}_{n1}^{(s)} \mathbf{L}_{n1}^{(s)\top}$ is positive semidefinite, its smallest eigenvalue is nonnegative. Moreover, each normalized Laplacian $\mathbf{L}_{n1}^{(t)}$ is symmetric and positive semidefinite, so

$$\sqrt{\lambda_k \left( \mathbf{L}_{n1}^{(t)} \mathbf{L}_{n1}^{(t)\top} \right)} = \lambda_k \left( \mathbf{L}_{n1}^{(t)} \right).$$

Thus, for all $t \in \{1, \ldots, T\}$,

$$\sigma_k \geq \lambda_k \left( \mathbf{L}_{n1}^{(t)} \right).$$

Applying the higher order Cheeger inequality yields

$$\phi_k(G^{(t)}) \leq \mathrm{poly}(k) \sqrt{\lambda_k \left( \mathbf{L}_{n1}^{(t)} \right)} \leq \mathrm{poly}(k) \sqrt{\sigma_k},$$

and therefore

$$\phi_k(\mathcal{G}) = \max_{t \in \{1, \ldots, T\}} \phi_k(G^{(t)}) \leq \mathrm{poly}(k) \sqrt{\sigma_k}.$$

For the reverse inequality, Weyl's inequality gives

$$\sigma_k \leq \sqrt{\lambda_k \left( \mathbf{L}_{n1}^{(t)} \mathbf{L}_{n1}^{(t)\top} \right) + \lambda_n \left( \sum_{s \neq t} \mathbf{L}_{n1}^{(s)} \mathbf{L}_{n1}^{(s)\top} \right)}.$$

Using

$$\lambda_n \left( \sum_{s \neq t} \mathbf{L}_{n1}^{(s)} \mathbf{L}_{n1}^{(s)\top} \right) = \|\mathcal{L}_{n1}^{-t}\|_2^2, \qquad \lambda_k \left( \mathbf{L}_{n1}^{(t)} \mathbf{L}_{n1}^{(t)\top} \right) = \lambda_k \left( \mathbf{L}_{n1}^{(t)} \right)^2,$$

we obtain

$$\sigma_k \leq \sqrt{\lambda_k\left(\mathbf{L}_{n1}^{(t)}\right)^2 + \|\mathcal{L}_{n1}^{-t}\|_2^2}.$$

Applying the higher order Cheeger inequality again gives

$$\lambda_k\left(\mathbf{L}_{n1}^{(t)}\right) \leq 2\phi_k(G^{(t)}),$$

and hence

$$\sigma_k \leq \sqrt{(2\phi_k(G^{(t)}))^2 + \|\mathcal{L}_{n1}^{-t}\|_2^2}.$$

Rearranging yields

$$\sigma_k^2 - \|\mathcal{L}_{n1}^{-t}\|_2^2 \leq (2\phi_k(G^{(t)}))^2.$$

Taking the minimum over $t$ on the left-hand side and the maximum on the right-hand side gives

$$\sigma_k^2 - \min_{t\in\{1,\ldots,T\}} \|\mathcal{L}_{n1}^{-t}\|_2^2 \leq \left(2 \max_{t\in\{1,\ldots,T\}} \phi_k(G^{(t)})\right)^2 = (2\phi_k(\mathcal{G}))^2,$$

which completes the proof. $\qquad\square$

## H. Stability Properties of the Anchor Embedding

Under the same assumptions as in Theorem 1, there exists a matrix $\mathbf{X} \in \mathbb{R}^{n\times d}$ such that

$$\|\hat{\mathbf{X}} - \mathbf{X}\|_2 = \mathcal{O}\left(\frac{1}{\rho^{1/2}n^{1/2}}\right) \quad \text{a.s.}$$

and the following stability property hold: if $\mathbf{z}_i = \mathbf{z}_j$, then $\mathbf{X}_{i:} = \mathbf{X}_{j:}$.

In particular, setting $\mathbf{X} = \tilde{\mathbf{X}}\mathbf{W}$ satisfies the desired stability properties. The result follows by combining a convergence statement for the empirical anchor embedding with an exact noise-free stability property.

**Corollary H.1.** *There exists* $\mathbf{W} \in \mathbb{O}((K-1)\times(K-1))$ *such that*

$$\|\hat{\mathbf{X}} - \tilde{\mathbf{X}}\mathbf{W}\|_2 = \mathcal{O}\left(\frac{1}{\rho^{1/2}n^{1/2}}\right) \quad \text{a.s.}$$

*Proof.* By Corollary E.5, Corollary E.10, and Corollary E.11,

$$
\begin{aligned}
\|\hat{\mathbf{X}} - \tilde{\mathbf{X}}\mathbf{W}\|_2 &= \|\mathbf{U}\boldsymbol{\Sigma}^{1/2} - \tilde{\mathbf{U}}\tilde{\boldsymbol{\Sigma}}^{1/2}\mathbf{W}\|_2 \\
&\leq \|(\mathbf{U} - \tilde{\mathbf{U}}\mathbf{W})\boldsymbol{\Sigma}^{1/2} + \tilde{\mathbf{U}}(\mathbf{W}\boldsymbol{\Sigma}^{1/2} - \tilde{\boldsymbol{\Sigma}}^{1/2}\mathbf{W})\|_2 \\
&\leq \|\mathbf{U} - \tilde{\mathbf{U}}\mathbf{W}\|_2\,\|\boldsymbol{\Sigma}^{1/2}\|_2 + \|\tilde{\mathbf{U}}\|_2\,\|\mathbf{W}\boldsymbol{\Sigma}^{1/2} - \tilde{\boldsymbol{\Sigma}}^{1/2}\mathbf{W}\|_2 \\
&\leq \sigma_n^{1/2}\|\mathbf{U} - \tilde{\mathbf{U}}\mathbf{W}\|_2 + \|\mathbf{W}\boldsymbol{\Sigma}^{1/2} - \tilde{\boldsymbol{\Sigma}}^{1/2}\mathbf{W}\|_F \\
&= \mathcal{O}\left(\frac{1}{\rho^{1/2}n^{1/2}}\right) \quad \text{a.s.}
\end{aligned}
$$

$\qquad\square$

**Corollary H.2.** *The noise-free anchor embedding* $\tilde{\mathbf{X}}$ *is constant within communities:*

$$\mathbf{z}_i = \mathbf{z}_j \;\Rightarrow\; \tilde{\mathbf{X}}_{i:} = \tilde{\mathbf{X}}_{j:}.$$

*Proof.* By Lemma E.3 and Corollary E.4, the left singular vectors $\tilde{\mathbf{U}}$ corresponding to the $K-1$ nontrivial components lie in the subspace $\mathcal{S}$. Hence each column of $\tilde{\mathbf{U}}$ is of the form $\mathcal{Z}\mathbf{q}$ for some $\mathbf{q} \in \mathbb{R}^K$, so $\tilde{\mathbf{U}}$ is piecewise constant across the rows indexed by the same community label. In particular, if $\mathbf{z}_i = \mathbf{z}_j$, then

$$\tilde{\mathbf{U}}_{i:} = \tilde{\mathbf{U}}_{j:}.$$

Since $\tilde{\mathbf{X}} = \tilde{\mathbf{U}}\tilde{\boldsymbol{\Sigma}}^{1/2}$, multiplying by the common matrix $\tilde{\boldsymbol{\Sigma}}^{1/2}$ preserves row equality, and therefore

$$\tilde{\mathbf{X}}_{i:} = \tilde{\mathbf{U}}_{i:}\tilde{\boldsymbol{\Sigma}}^{1/2} = \tilde{\mathbf{U}}_{j:}\tilde{\boldsymbol{\Sigma}}^{1/2} = \tilde{\mathbf{X}}_{j:}.$$

$\qquad\square$

# I. Details of Synthetic Datasets

We consider two synthetic dynamic network settings generated from a dynamic stochastic block model observed over $T = 3$ time steps. The number of communities varies over time and is given by

$$K^{(1)} = 3, \qquad K^{(2)} = 2, \qquad K^{(3)} = 2.$$

Although the number and composition of communities change over time, we assume that this evolution can be explained by $K = 3$ distinct community trajectories, each describing how a node's community membership evolves over time.

The temporal evolution is governed by deterministic mappings $\{c^{(t)}\}_{t=1}^3$, where $c^{(t)}(k)$ denotes the community index at time $t$ associated with the $k$-th trajectory. These mappings are given by

$$c^{(1)}(1) = 1, \quad c^{(1)}(2) = 2, \quad c^{(1)}(3) = 3,$$
$$c^{(2)}(1) = 1, \quad c^{(2)}(2) = 1, \quad c^{(2)}(3) = 2,$$
$$c^{(3)}(1) = 1, \quad c^{(3)}(2) = 2, \quad c^{(3)}(3) = 2.$$

Equivalently, the correspondence between latent trajectories and observable communities at each time point can be described as follows:

- At time $t = 1$, all trajectories correspond to distinct communities: trajectory $k$ forms community $k$ for $k = 1, 2, 3$.

- At time $t = 2$, trajectories 1 and 2 merge and jointly form community 1, while trajectory 3 alone forms community 2.

- At time $t = 3$, trajectory 1 alone forms community 1, while trajectories 2 and 3 merge and jointly form community 2.

The two experimental settings differ only in the distribution $\boldsymbol{\pi}$ over latent trajectories:

$$\begin{cases} \text{Experiment 1:} & \boldsymbol{\pi} = (0.3,\ 0.4,\ 0.3), \\ \text{Experiment 2:} & \boldsymbol{\pi} = (0.4,\ 0.5,\ 0.1). \end{cases}$$

The second configuration introduces a substantial imbalance in community sizes, resulting in markedly increased degree heterogeneity across nodes. This setting is designed to assess the robustness of Laplacian based dynamic embeddings in comparison to adjacency based methods under heterogeneous degree regimes.

# J. Additional Experimental Details

### J.1. Baseline Implementations and Hyperparameters

For the temporal deep learning baselines, we used the official implementations and followed the hyperparameter settings and evaluation protocols recommended in the corresponding repositories or papers. Our goal was to compare ULSE against standard, reproducible baseline configurations rather than introduce additional dataset specific tuning choices. This choice is particularly important because these baselines optimize predictive objectives, such as link prediction, whereas ULSE is an unsupervised spectral embedding method designed to recover a stable latent geometry. Extensive per dataset tuning of the baselines could therefore introduce task and dataset dependent variability that is not directly tied to the stability properties studied in this paper.

In our DyGLib based implementation, the shared training configuration used batch size 32, Adam optimizer with learning rate $10^{-3}$ and weight decay $5 \times 10^{-4}$, dropout 0.1, two layers, four attention heads, time-feature dimension 32, uniform temporal-neighbor sampling with five neighbors, time scaling factor 10.0, random negative sampling, and 50 training epochs with patience 5. For DyGFormer, we used patch size 16, channel embedding dimension 12, and maximum input sequence length 10. No validation or test split was used in this stage, since the trained models were used to extract node embeddings for downstream clustering rather than to tune predictive performance.

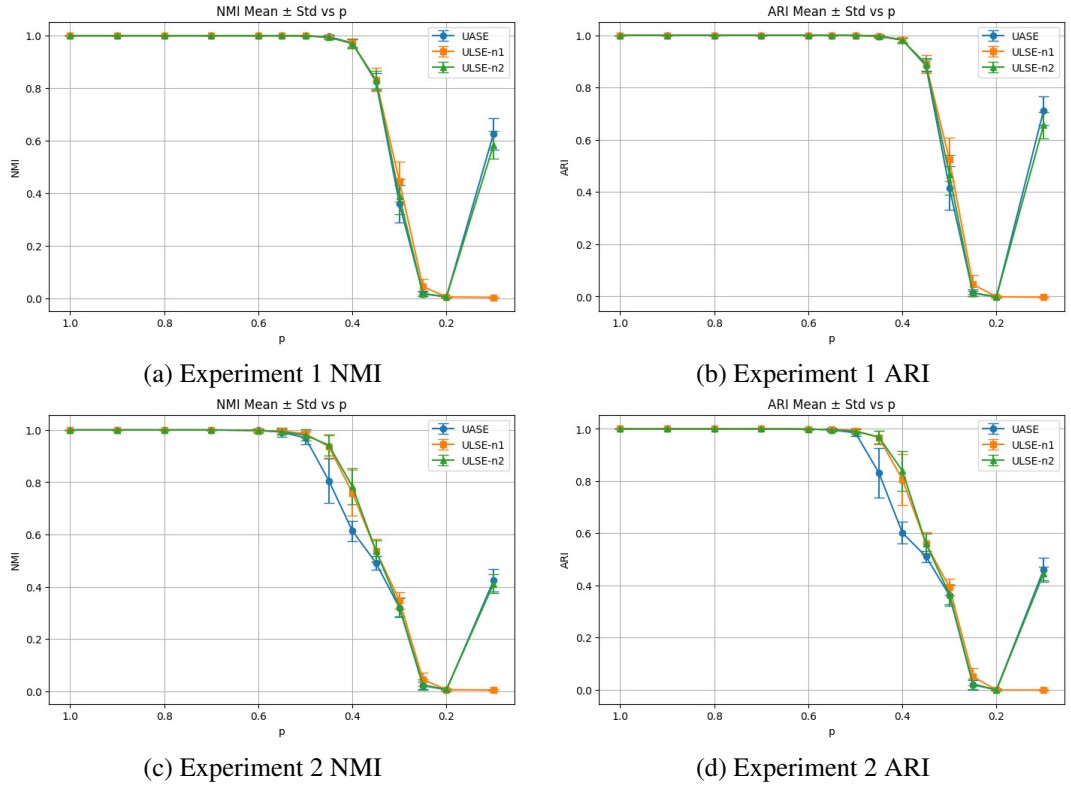

(a) Experiment 1 NMI          (b) Experiment 1 ARI

(c) Experiment 2 NMI          (d) Experiment 2 ARI

*Figure 3.* Clustering performance (NMI and ARI) for Experiments 1 and 2 as the intra-community probability $p$ varies.

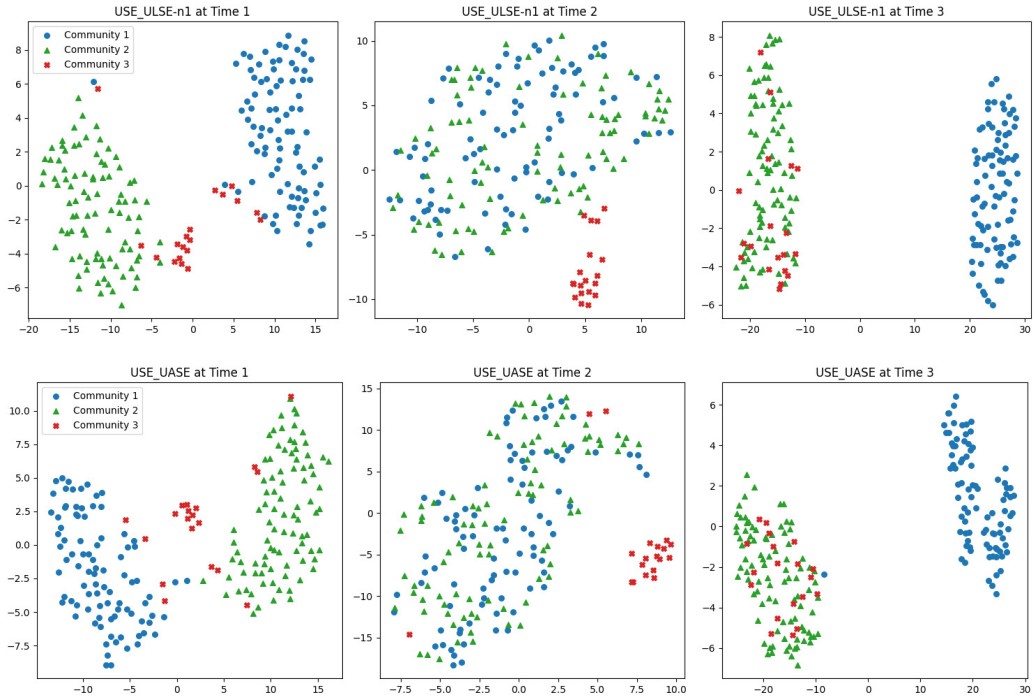

*Figure 4.* t-SNE visualizations of dynamic embeddings for the second synthetic dataset with $p = 0.4$. Top: ULSE-n1. Bottom: UASE.

## K. Further Comparison among ULSE-n1, ULSE-n2 and UASE

In the second synthetic dataset, the imbalance in community size distributions leads to substantially lower degrees for nodes in the smallest community. As reported in Table 1 of the main paper, this degree heterogeneity has a markedly stronger negative impact on UASE than on ULSE.

To further examine this effect, Figure 3 reports the performance of ULSE-n1, ULSE-n2, and UASE in terms of NMI and ARI as the intra-community connection probability $p$ varies, with the inter-community probability fixed at $q = 0.2$. For the second synthetic dataset, UASE performance degrades sharply in the range $p \in [0.4, 0.45]$, while both ULSE-n1 and ULSE-n2 remain stable. This behavior explains the superior performance of ULSE observed in the main experimental results.

The statistical significance of this performance gap is summarized in Table 3, which reports one-sided $p$-values comparing ULSE against UASE for both NMI and ARI on the second synthetic dataset.

*Table 3.* One-sided $p$-values comparing ULSE and UASE on the second synthetic dataset

| Method | Metric | $p = 0.4$ | $p = 0.45$ |
|--------|--------|-----------|------------|
| ULSE-n1 | NMI | $1.535 \times 10^{-7}$ | $4.899 \times 10^{-7}$ |
|         | ARI | $2.441 \times 10^{-9}$ | $1.952 \times 10^{-6}$ |
| ULSE-n2 | NMI | $7.8 \times 10^{-11}$ | $2.347 \times 10^{-7}$ |
|         | ARI | $< 1 \times 10^{-12}$ | $1.552 \times 10^{-6}$ |

The root cause of this discrepancy lies in the design of UASE, which is known to suppress information from low-degree nodes by treating it as noise (Krzakala et al., 2013). In contrast, ULSE-n1 incorporates degree normalization and leverages smaller singular values of the unfolded normalized Laplacian, thereby preserving informative structure associated with low-degree nodes.

For additional qualitative insight against UASE, Figure 4 shows t-SNE (van der Maaten & Hinton, 2008) projections of the learned embeddings for the second synthetic dataset with $p = 0.4$. ULSE-n1 produces well separated and temporally consistent clusters, whereas UASE misplaces nodes from the smallest community (shown as red crosses), scattering them across the embedding space. This misalignment directly contributes to the degraded clustering performance of UASE in this regime.

However, ULSE-n1 may exhibit degradation in low-$p$ regimes when the graph becomes increasingly heterophilous ($p < q$). In this setting, informative community structure in the normalized Laplacian shifts toward larger eigenvalues. Since ULSE-n1 relies on the smaller singular values of the unfolded normalized Laplacian, the resulting embeddings become less informative. In contrast, UASE and ULSE-n2 retain informative spectral structure in their respective operators and therefore maintain stronger performance in this regime.

To further clarify the difference between ULSE-n1 and ULSE-n2, we examine regimes in which aggregated normalization is beneficial. The two variants differ in how degree normalization is handled across time. ULSE-n1 uses per snapshot normalization and is therefore closely tied to the spectrum of each individual normalized Laplacian. This is effective when degree distributions are relatively stable and the informative structure is captured by the smaller nontrivial singular directions. In contrast, ULSE-n2 uses a time-aggregated left normalization, which stabilizes node-wise scaling across snapshots. We therefore expect ULSE-n2 to be preferable when degree distributions vary substantially over time or when the community structure is heterophilous.

To quantify longitudinal stability, we compute community-level displacement statistics. For each community whose connectivity profile is preserved between two snapshots, we compute the average Euclidean displacement of its embedded nodes across the corresponding time points. We define the displacement ratio as

$$\text{DispRatio}(k) = \frac{\text{average displacement of community } k \text{ when its connectivity profile is unchanged}}{\text{average displacement of community } k \text{ when its connectivity profile changed}}.$$

A smaller displacement ratio indicates stronger longitudinal stability.

Figure 5 compares ULSE-n1 and ULSE-n2 as the intra-community probability $p$ varies from 0.1 to 1.0, with the inter-community probability fixed at $q = 0.5$. When $p > q$, the main source of instability is temporal degree variation, and

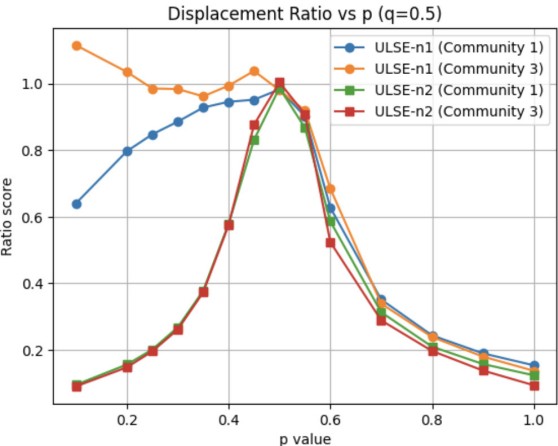

*Figure 5.* Displacement ratios for ULSE-n1 and ULSE-n2 as the intra-community probability $p$ varies, with inter-community probability fixed at $q = 0.5$. Smaller values indicate stronger longitudinal stability. We report results for communities whose connectivity profiles are preserved across the relevant time points.

ULSE-n2 yields slightly smaller displacement ratios than ULSE-n1. When $p < q$, the graph becomes heterophilous and the informative structure shifts away from the smaller singular directions used by ULSE-n1. In this regime, ULSE-n2 achieves substantially smaller displacement ratios, indicating stronger longitudinal stability.

The embedding visualizations in Figure 6 provide a qualitative explanation for this behavior. In the heterophilous regime $p < q$, ULSE-n1 fails to clearly distinguish communities because the smaller singular directions of the unfolded normalized Laplacian become less informative. By contrast, ULSE-n2 retains the relevant structural information through its aggregated normalization and produces embeddings with clearer community separation. These results support the use of ULSE-n2 in settings with strong temporal degree variability or heterophilous community structure.

## L. Stability Results Including UASE

For completeness, we report the full set of stability experiments presented in the main paper, now including a direct comparison with UASE in Figure 7. All embeddings (UASE, ULSE-n1, and ULSE-n2) are three-dimensional. The figure displays two coordinates corresponding to the two largest singular values for UASE and ULSE-n2, and the two smallest nontrivial singular values for ULSE-n1.

Across the evaluated synthetic settings, all three methods recover the expected temporal evolution of communities and exhibit consistent alignment across snapshots. In particular, UASE also satisfies the stability conditions in this controlled regime, consistent with its theoretical guarantees under dynamic latent position models. This experiment highlights that ULSE preserves stability properties comparable to UASE while extending them to Laplacian based operators.

## M. Stability Analysis of Deep Learning Methods

We evaluate whether representative deep learning models for temporal graphs satisfy the stability conditions studied in this thesis using the synthetic dynamic networks described earlier. For qualitative assessment, we apply t-SNE (van der Maaten & Hinton, 2008) to project learned node embeddings into two dimensions. Although t-SNE does not preserve global geometry, it is sufficient to reveal violations of cross-sectional stability and to visually assess whether community structure remains coherent across time.

As stable baselines, we include t-SNE visualizations of ULSE-n1 and ULSE-n2. We compare these against embeddings produced by four widely used deep learning approaches for temporal networks: JODIE (Kumar et al., 2019), DyRep (Trivedi et al., 2019), TGN (Rossi et al., 2020), and DyGFormer (Yu et al., 2023). All models are trained and evaluated using their standard, publicly documented protocols.

Figures 8 and 9 show that ULSE-n1 and ULSE-n2 yield embeddings with clearly separated communities that remain aligned across time steps, reflecting both cross-sectional and longitudinal stability. In contrast, the deep learning methods

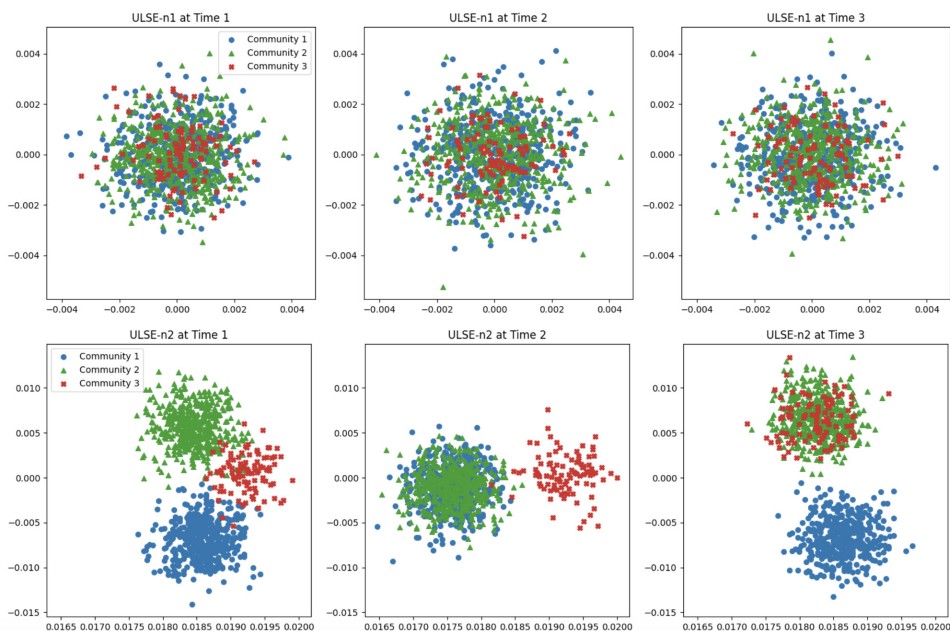

*Figure 6.* Embedding visualizations comparing ULSE-n1 and ULSE-n2 in a regime where $p < q$. ULSE-n1 fails to clearly separate the communities, whereas ULSE-n2 better preserves community separation and temporal alignment.

exhibit substantial instability: community clusters fragment, overlap, or drift across snapshots, indicating that the learned embeddings are not comparable over time.

These results are consistent with the design objectives of existing deep temporal graph models. Such methods are primarily optimized for downstream predictive tasks (e.g., link prediction or event forecasting) and do not impose geometric constraints that enforce temporal alignment or invariance. Consequently, their embeddings generally fail to satisfy the stability properties required for principled longitudinal analysis.

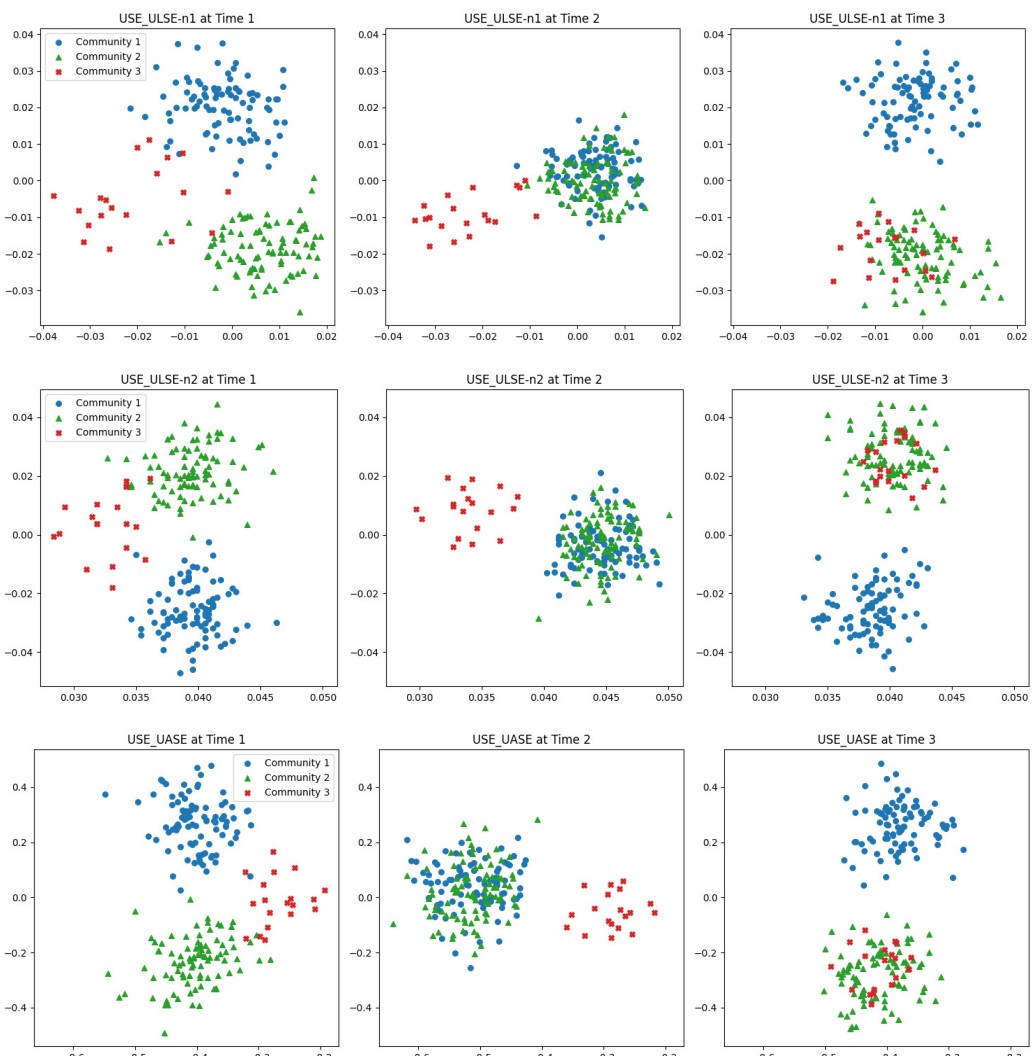

*Figure 7.* Three-dimensional dynamic embeddings on the synthetic dataset. Top: ULSE-n1. Middle: ULSE-n2. Bottom: UASE.

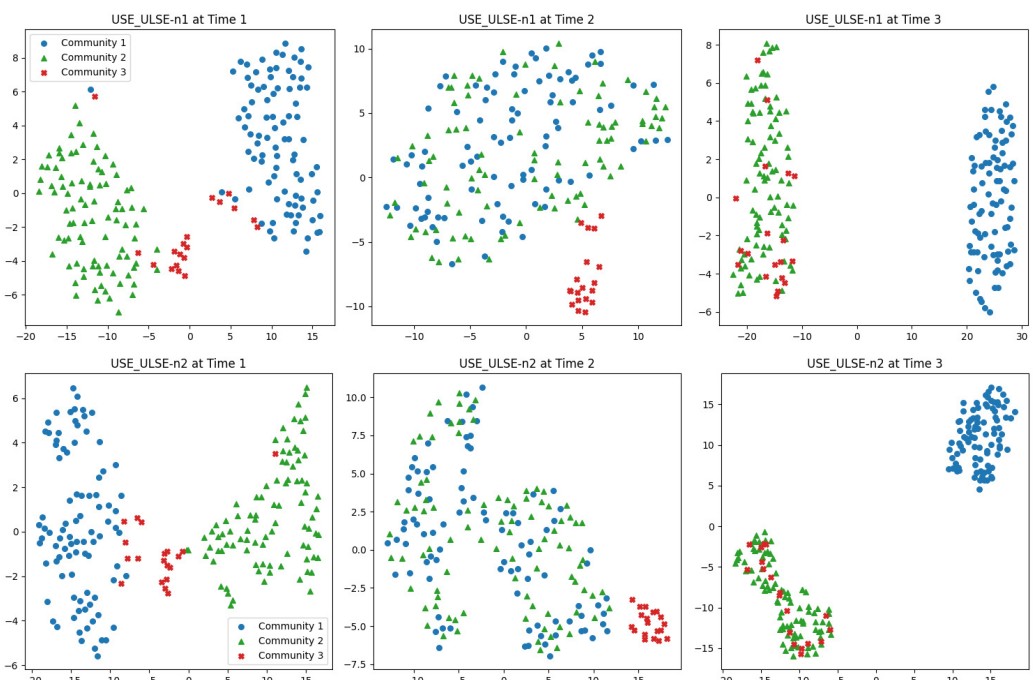

*Figure 8.* t-SNE projections of dynamic embeddings produced by ULSE. Top: ULSE-n1. Bottom: ULSE-n2.

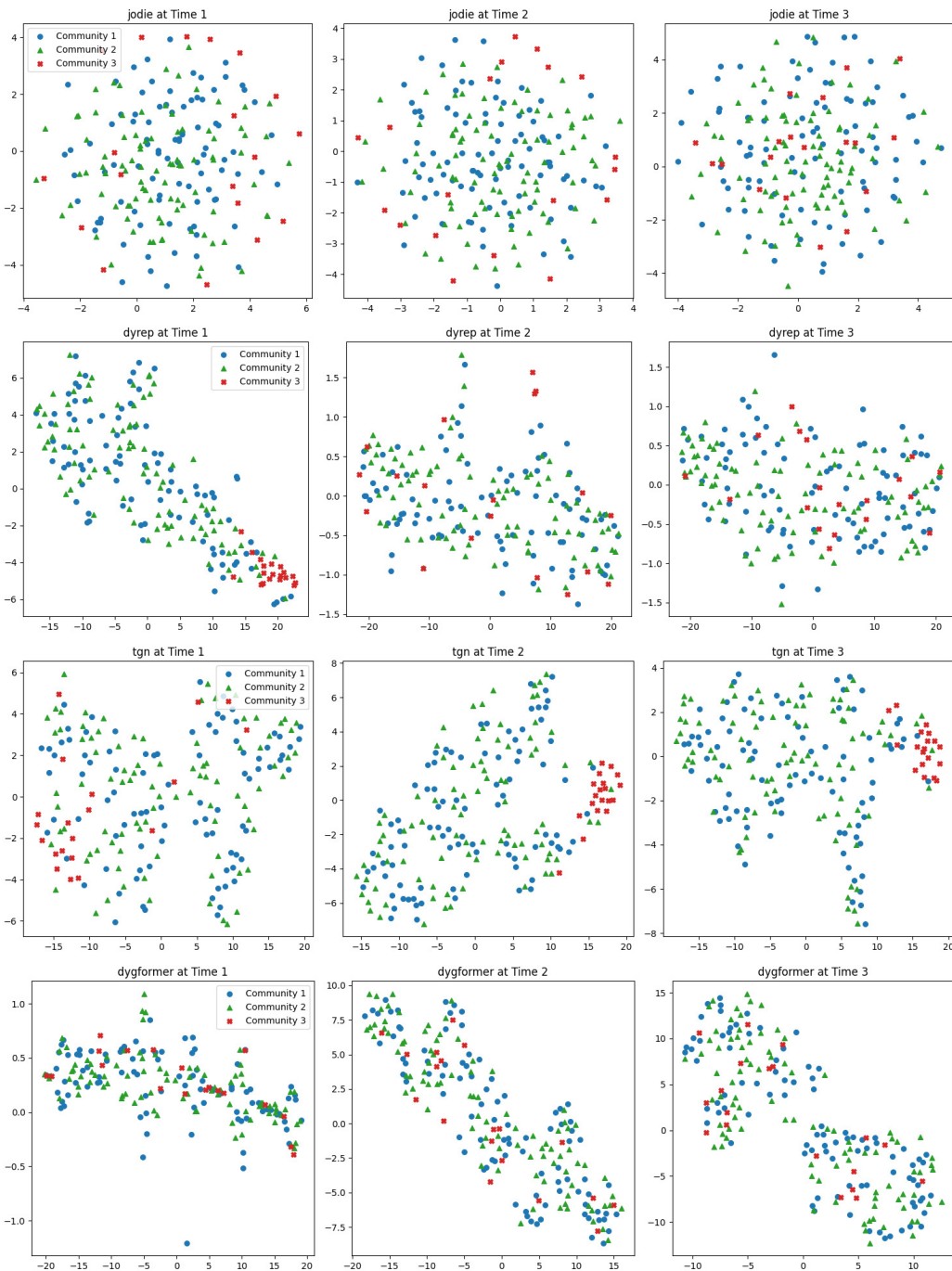

*Figure 9.* t-SNE projections of dynamic embeddings learned by deep temporal graph models. From top to bottom: JODIE, DyRep, TGN, and DyGFormer.

