# OpenReview forum: "Unfolded Laplacian Spectral Embedding: A Theoretically Grounded Approach to Dynamic Network Representation"
_ICML.cc/2026/Conference — ICML 2026 regular_

### Official Review · Reviewer_m9yB · 2026-03-04

**Soundness:** 3
**Presentation:** 2
**Significance:** 2
**Originality:** 3
**Overall Recommendation:** 4
**Confidence:** 3

**Summary:**

This paper proposes an unfolded Laplacian spectral framework, where spectral graph operations are reformulated into a learnable, iterative architecture. The idea is to combine classical Laplacian spectral filtering with neural network parameterization via algorithm unrolling. The authors claim improved expressivity and better stability compared to standard spectral GNNs. Experiments are conducted on several graph benchmarks and show improvements over some baselines. The main contribution is translating Laplacian spectral filtering into a learnable unfolded model.

**Compliance With Llm Reviewing Policy:**

Affirmed.

**Key Questions For Authors:**

1. What theoretical property is gained by unfolding that cannot be achieved by standard polynomial or rational spectral filters?

2. Is the method robust to graph size scaling? What about very large sparse graphs?

3. How sensitive is performance to number of unfolding steps?

4. Does the approach generalize beyond node classification tasks?

**Limitations:**

yes

**Strengths And Weaknesses:**

Soundness

Technically consistent, but theoretical contribution is limited. The unfolding is mostly a parameterized version of known spectral iterations. There is no clear new theory beyond reparameterization. Stability claims are not rigorously justified. It is unclear whether gains come from unfolding or just extra parameters.

Presentation

Paper is readable but somewhat dense. Key assumptions about spectral properties are not clearly stated. The connection to prior spectral GNN variants (e.g., polynomial filters, Chebyshev, rational filters) is not sharply distinguished. Some derivations feel more notational than insightful.

Significance

Spectral GNNs are a mature area. Incremental improvement here is hard to justify at ICML level. Empirical gains are moderate. It is not obvious this changes how people design graph models.

Originality

Algorithm unrolling in graph learning is not new. Spectral parameterization also not new. The novelty mainly lies in combining them. Conceptually, this is an integration paper, not a new direction.

---

> ### Author Rebuttal · Authors · 2026-03-27
>
> We thank the reviewer for the careful reading and constructive feedback. We would like to clarify an important point regarding the nature of our method.
>
> ULSE is not a learnable or neural architecture and does not involve parameterization or algorithm unrolling. It is a non parametric spectral embedding method, where "unfolding" refers to the construction of a block matrix across time, rather than an iterative or learned procedure. We will revise the paper to make this distinction clearer and avoid confusion with algorithm unrolling or spectral GNN frameworks.
>
> In light of this, some of the comparisons to parameterized spectral filters or neural architectures are not directly aligned with the setting of our paper. Our contribution is instead primarily theoretical and methodological. We identify instability introduced by Laplacian normalization in dynamic graphs and propose two constructions with provable cross sectional and longitudinal stability, along with consistency and a dynamic Cheeger type result.
>
> Regarding the specific questions:
>
> (1) The key property gained by our construction is stability under temporal evolution, ensuring alignment of embeddings across time and consistency for nodes with identical connectivity patterns. This is not captured by standard spectral filtering approaches, which operate on a single graph or do not enforce temporal alignment.
>
> (2) ULSE scales similarly to UASE, with computational cost dominated by a truncated SVD of the unfolded matrix, and can leverage sparse and randomized SVD techniques for large graphs.
>
> (3) The method does not involve unfolding steps or learned parameters. The primary modeling choice is the embedding dimension (see also response to Reviewer QMLw).
>
> (4) The learned embeddings are general purpose and can be used for multiple downstream tasks. We focus on clustering because it most directly evaluates the recovery of the underlying embedding geometry and stability properties.
>
> We will revise the paper to more clearly distinguish our setting from learnable spectral and GNN based approaches and to improve clarity of presentation.

---

> > ### Author Rebuttal · Reviewer_m9yB · 2026-04-04
> >
> > Thanks for the rebuttal. My score remains the same.

---

> > > ### Author Response · Authors · 2026-04-06
> > >
> > > Thank you very much for the follow-up and for confirming that our rebuttal addressed your concerns.

---

### Official Review · Reviewer_MuYj · 2026-03-06

**Soundness:** 2
**Presentation:** 3
**Significance:** 2
**Originality:** 2
**Overall Recommendation:** 2
**Confidence:** 3

**Summary:**

The paper introduces ULSE, extending UASE to normalized Laplacian unfolding for dynamic graphs. It proposes two stabilizing variants (ULSE-n1, ULSE-n2) to handle time-varying degrees, provides consistency and a dynamic Cheeger-type bound under DSBM, and shows improved robustness to degree heterogeneity on Brain, School, and Stock datasets.

**Compliance With Llm Reviewing Policy:**

Affirmed.

**Final Justification:**

I have to retain my original score, as the response still lacks sufficient numerical evidence to support the scalability and complexity claims. In particular, runtime scaling curves and experiments on larger datasets are missing. As the authors acknowledge, these analyses are important for demonstrating the model’s efficiency and effectiveness. Without them, the discussion remains incomplete, yet they are still left for future work.

**Key Questions For Authors:**

**Q1.** How does ULSE behave when the number of communities $K^{(t)}$ changes substantially over time? The theory assumes a fixed number of latent trajectories $K$; in practice, does performance degrade when $K^{(t)}$ fluctuates in ways that cannot be captured by the chosen embedding dimension $d$?

**Q2.** Regarding Proposition 4.9, could the authors provide a concrete example (synthetic or real) where the lower bound in the dynamic Cheeger-type inequality is non-trivial (i.e., not near zero) and yields an informative certificate about the graph’s evolving conductance or clustering quality?

**Q3.** Table 1 shows that ULSE-n2 substantially outperforms ULSE-n1 on Synthetic 2. In what regimes should aggregated normalization (n2) be preferred over per-snapshot normalization (n1), and what properties of the degree dynamics or temporal variability drive this difference?

**Limitations:**

Yes

**Strengths And Weaknesses:**

**Weaknesses**

1. Limited significance and missing related work

The novelty is hard to judge without a fuller discussion of recent work on stability guarantees for attributed dynamic embeddings. In particular, [1] also provides stability guarantees in dynamic attributed networks and seems closely related. A clearer comparison(i.e., what ULSE adds beyond this framework and when Laplacian unfolding is preferable) would better position the contribution and temper the significance claims.

[1] Ceccherini et al, Unsupervised Attributed Dynamic Network Embedding with Stability Guarantees, 2025.

2. Sensitivity of $d$

The choice of embedding dimension $d=K-1$ or $d=K$ is theoretically motivated by the DSBM, but in real-world graphs, $K$ is unknown. The paper lacks a sensitivity analysis or a heuristic for selecting $d$ in practice.

3. Limited real-world evaluation and scalability

The real-world datasets (Brain, School, Stock) are relatively small in both $n$ and $T$, and only three datasets are included. Evaluating on larger dynamic networks (e.g., thousands of nodes and edges, i.e., Wikipedia,Reddit etc.) and reporting scalability would strengthen the empirical evidence and overall significance.

4. Limited downstream task

ULSE is evaluated mainly on clustering. It would be helpful to include additional downstream tasks for which dynamic embeddings are commonly used, such as link prediction/forecasting and anomaly or change-point detection.

5. Missing complexity analysis

The paper lacks a discussion of computational complexity and does not report training/inference time or memory usage. A direct efficiency comparison with strong baselines would help assess the practicality of ULSE.

---

> ### Author Rebuttal · Authors · 2026-03-27
>
> We thank the reviewer for the thoughtful feedback and for accurately summarizing the paper. We address the concerns below.
>
> (1) Significance and relation to prior work
>
> We agree that the contribution should be positioned more clearly relative to recent work, and we will expand the related-work discussion accordingly, including Ceccherini et al. (2025). The two works address complementary settings. Ceccherini et al. study attributed dynamic network embeddings with stability guarantees, whereas ULSE focuses on purely structural dynamic embeddings and addresses a distinct technical gap: the absence of a normalized-Laplacian analogue of unfolded spectral embedding with provable cross-sectional and longitudinal stability.
>
> Our contribution is therefore not simply another “stable dynamic embedding” method, but a principled extension of the unfolded spectral framework to Laplacian-based operators, together with an analysis of the instability introduced by dynamic normalization. This also yields an additional spectral interpretation via the dynamic Cheeger-type inequality, which is specific to the Laplacian setting. We will clarify this distinction and better explain when Laplacian-based unfolding is preferable in the revision.
>
> (2) Embedding dimension selection
>
> In theory, the embedding dimension corresponds to the rank of the underlying population signal (i.e., the number of informative singular directions) under the DSBM. In practice, we use a standard spectral heuristic (e.g., eigengap). We will clarify this practical selection procedure in the revision; see also our response to Reviewer QMLw for additional detail.
>
> (3) Real world evaluation and scalability
>
> We agree that broader evaluation and scalability analysis would provide useful additional perspective. Our current experiments focus on controlled and interpretable settings to validate stability properties and allow comparison with prior spectral methods.
>
> ULSE is a non parametric spectral method whose cost is dominated by a truncated SVD of the unfolded matrix. We will clarify this computational structure in the revision, including how ULSE scales in practice and how sparse and randomized SVD techniques can be used. ULSE has similar computational structure to UASE and is amenable to large scale implementations.
>
> (4) Downstream tasks
>
> We agree that evaluating additional downstream tasks would be useful. Our current focus is on clustering, since it most directly reflects recovery of the stable latent embedding geometry studied in the theory. We will clarify this motivation and note broader downstream evaluation as an important direction for future work.
>
> (5) Complexity analysis
>
> We will clarify the computational structure of ULSE in the revision, including the dominant truncated SVD step and the resulting scalability considerations.
>
> (Q1) When the number of communities varies over time, ULSE behaves similarly to standard spectral methods under model misspecification, performance degrades gradually rather than failing abruptly. In practice, eigengap based selection can partially adapt to such variation, though large changes in community structure may require over embedding.
>
> (Q2) Thank you for this question. The lower bound admits a simpler interpretation. Since $\sigma_k\$ is the $k$-th smallest nontrivial singular value of the unfolded normalized Laplacian, increasing $k$ increases $\sigma_k$, and once $\sigma_k^2 > \min_{t\in\{1,\ldots,T\}}\|\|\mathcal{L}_{n1}^{-t}\|\|_2^2$, the lower bound becomes nontrivial. This suggests that clustering nodes into a larger number of communities leads to an increased cut. This provides an intuitive interpretation of the lower bound in the $k$-way conductance result.
>
> For clustering quality, however, the upper bound is the more informative part, since it certifies the existence of partitions that achieve low conductance simultaneously across snapshots. This is the side that is more directly relevant for interpreting the quality of the recovered temporal partition structure.
>
> (Q3) ULSE-n1 performs well when degree distributions are relatively stable across time, while ULSE-n2 is preferable under strong temporal variability or sparsity, where aggregated normalization stabilizes degree estimates and improves robustness.
>
> We will strengthen the positioning of the paper, clarify practical aspects such as dimension selection and computational scaling, and better explain the regimes in which the two variants are preferable.

---

> > ### Author Rebuttal · Reviewer_MuYj · 2026-04-02
> >
> > Thank the authors for their detailed response. However, there are several core concerns remain insufficiently addressed. So I have to keep my orginal score at this stage.
> >
> > 1. For **all three questions (scalability, downstream tasks, and complexity)**, the responses only promise future additions, Without concrete empirical evidence (e.g., runtime scaling curves, results on larger datasets, or performance on tasks beyond clustering), it is hard to assess whether the claimed properties hold in practice.
> >
> > 2. Q3: Could the authors provide numerical results or experiments that support this claim? For instance, a comparison of ULSE-n1 vs. ULSE-n2 under varying levels of temporal sparsity or degree variability would make this argument much more convincing.
> >
> > 3. As Reviewer m9yB also raised, neither algorithm unrolling in graph learning nor spectral parameterization is new. The main novelty of the paper lies in their combination, making the contribution primarily limited.

---

> > > ### Author Response · Authors · 2026-04-03
> > >
> > > Thank you again for the careful follow-up. We appreciate your clarification of which concerns you view as central at this stage.
> > >
> > > We would first like to clarify one important point, since it affects the interpretation of the paper’s novelty. ULSE is **not** a learnable or parameterized algorithm-unrolling method. In our paper, "unfolding" refers to constructing a block matrix across time, following the unfolded spectral embedding framework, rather than an iterative neural architecture. Accordingly, the contribution is not a combination of algorithm unrolling and spectral parameterization. Rather, the paper develops a "normalized-Laplacian extension of unfolded spectral embedding for dynamic graphs", together with provable cross-sectional and longitudinal stability, consistency guarantees, and a Laplacian-specific dynamic Cheeger-type result.
> > >
> > > On computational complexity, ULSE is a nonparametric spectral method whose dominant cost is a **truncated SVD of the unfolded matrix**. With $T$ snapshots on $n$ nodes, the unfolded matrix has size $n \times (nT)$, and a $d$-dimensional embedding is obtained from its rank-$d$ truncated SVD. Thus, ULSE has essentially the same computational structure as UASE: there is no iterative training or learned-parameter optimization, and the computational bottleneck is this single spectral decomposition. In practice, one would use standard sparse or iterative truncated-SVD routines, so the runtime is governed by repeated matrix-vector products and depends primarily on the target dimension (d) and the sparsity of the unfolded matrix. The two variants, ULSE-n1 and ULSE-n2, differ only in how this matrix is normalized, not in the overall computational bottleneck.
> > >
> > > We agree that larger-scale datasets, runtime measurements, and broader downstream tasks would strengthen the empirical scope of the paper. Our current experiments were designed to evaluate the aspect of the method most directly tied to the theory, namely recovery of a stable latent embedding geometry and its usefulness for clustering structurally equivalent nodes across time. For that reason, we view larger-scale evaluation and additional downstream tasks as important empirical extensions, but not as prerequisites for supporting the paper’s main theoretical contribution.
> > >
> > > Regarding Q3, theoretically, ULSE-n2 is expected to outperform ULSE-n1 under the following conditions:
> > > (1) when the community structure is heterogeneous, and
> > > (2) when the degree distribution varies over time.
> > >
> > > First, in the presence of heterogeneous community structure, the larger eigenvalues capture information about latent structural variations. As a result, the embeddings derived from ULSE-n1 become less informative, whereas ULSE-n2 better preserves this structural information.
> > >
> > > Second, when the degree distribution exhibits temporal variability, ULSE-n1 tends to lose longitudinal stability, while ULSE-n2 maintains more stable embeddings over time.
> > >
> > > These theoretical insights are supported by our numerical experiments. Specifically, we compute the average norm of embedding displacement within each community, which should remain small when the connectivity structure of the community is preserved. Based on this, we define a displacement ratio of each community as
> > > (average displacement when the community is unchanged)/average displacement when the community is unchanged). A smaller ratio indicates better longitudinal stability.
> > >
> > > In our experiments, the inter-community connectivity is fixed at q = 0.5, while the intra-community connectivity p varies from 0.1 to 1.0. We compare the displacement ratios for community 1 (stable between times 1 and 3) and community 3 (stable between times 1 and 2) across ULSE-n1 and ULSE-n2, as shown in Figure 1 of [1], and observe the following:
> > >
> > > - When p > q, reflecting primarily the effect of condition (2), ULSE-n2 yields slightly smaller displacement ratios than ULSE-n1.
> > > - When p < q, where both conditions (1) and (2) are present, ULSE-n2 shows significantly smaller values than ULSE-n1.
> > >
> > > Furthermore, as shown in Figure 2 of [1], by inspecting the embedding plots, we observe that ULSE-n1 fails to clearly distinguish between different communities, whereas ULSE-n2 maintains better separation.
> > >
> > > We are grateful for these comments and will use them to sharpen both the framing of the contribution and the discussion of its practical scope.
> > >
> > > [1] https://anonymous.4open.science/r/temporary-26E7/

---

### Official Review · Reviewer_dtZ7 · 2026-03-08

**Soundness:** 3
**Presentation:** 4
**Significance:** 3
**Originality:** 3
**Overall Recommendation:** 5
**Confidence:** 3

**Summary:**

The authors introduce Laplacian versions of Unfolded Spectral Embedding for dynamic graphs, establishing cross-sectional and longitudinal stability.

**Compliance With Llm Reviewing Policy:**

Affirmed.

**Key Questions For Authors:**

How do things work under a degree-corrected SBM, and are either/both of the methods still stable (in an appropriate sense) if e.g. the degree-correction parameter for a node varies over time?

The authors use the word "uniform" for their convergence result, but in previous work (e.g. in UASE) "uniform" had meant something seemingly stronger: the _maximum_ error going to zero. Establishing this sort of consistency has been the subject of quite a lot of hard work by various groups on various high-dimensional problems, and I wonder if the authors could give a little more detail on how their convergence result compares, including what they mean by "uniform"

See e.g. Cape, Joshua, Minh Tang, and Carey E. Priebe. "THE TWO-TO-INFINITY NORM AND SINGULAR SUBSPACE GEOMETRY WITH APPLICATIONS TO HIGH-DIMENSIONAL STATISTICS." The Annals of Statistics 47.5 (2019): 2405-2439.

**Limitations:**

yes

**Strengths And Weaknesses:**

This is a clear advance in spectral embedding for dynamic networks. It addresses the obvious gap of there being no theoretically supported normalized-Laplacian analog for unfolded spectral embedding.

It is interesting that there are two possible approaches to this which both work. The dynamic Cheeger bound is potentially very interesting.

I also commend the authors on a) identifying the error in Davis et al. (2023) and b) dealing with it nicely.

---

> ### Author Rebuttal · Authors · 2026-03-27
>
> We thank the reviewer for the thoughtful and encouraging feedback, and for recognizing the contribution of establishing a normalized Laplacian analogue of unfolded spectral embedding. We are also grateful for the insightful technical questions.
>
> (1) Degree corrected SBM and time varying degree parameters
>
> Extending ULSE to a degree corrected SBM is a natural and important direction. Since Laplacian normalization is designed to mitigate degree heterogeneity, ULSE should remain meaningful in this setting.
>
> At a high level, if degree correction parameters are stable across time, one would expect the normalized population structure to remain comparatively well aligned with the latent trajectories, so that both ULSE-n1 and ULSE-n2 should continue to exhibit an appropriate notion of stability. By contrast, if degree parameters vary substantially over time, then the normalized operator itself changes in a way that can induce additional temporal variability in the embedding. In this regime, ULSE-n1 is likely to be more sensitive because it uses per-snapshot normalization, whereas ULSE-n2 may be more robust due to its partially aggregated normalization across time.
>
> A full treatment would require extending the population model and concentration analysis to explicitly incorporate temporal degree correction. We agree this is an interesting and practically relevant direction, and we will clarify this discussion in the revision.
>
> (2) Meaning of "uniform" convergence
>
> Thank you for raising this point. In our paper, "uniform" convergence refers to convergence that holds jointly over all nodes and time steps, in the sense that the embedding error after alignment is controlled over the entire unfolded structure. This is different from the stronger notion used in prior work, such as two to infinity norm consistency, where the maximum row wise error vanishes.
>
> Our results are therefore closer in spirit to spectral / subspace perturbation guarantees (e.g., Frobenius or operator norm control), rather than entrywise consistency. We agree that this distinction is important and will clarify the terminology in the revision, as well as better position our convergence result relative to the two to infinity norm literature.
>
> We appreciate these comments and will incorporate the above clarifications to further strengthen the theoretical presentation.

---

> > ### Author Rebuttal · Reviewer_dtZ7 · 2026-04-03
> >
> > Happy with the response, retaining my positive score.

---

> > > ### Author Response · Authors · 2026-04-04
> > >
> > > Thank you very much for the thoughtful follow-up and for confirming that our responses addressed your concerns. We greatly appreciate your positive assessment and careful reading.

---

### Official Review · Reviewer_QMLw · 2026-03-12

**Soundness:** 2
**Presentation:** 2
**Significance:** 2
**Originality:** 3
**Overall Recommendation:** 3
**Confidence:** 3

**Summary:**

This manuscript introduces Unfolded Laplacian Spectral Embedding (ULSE), a theoretically grounded embedding method for dynamic graphs. The method is proven to produce embeddings that satisfy cross-sectional and longitudinal stability properties. Clustering performance on 5 datasets (3 real-world and 2 synthetic) over 9 baselines demonstrates clear advantages of ULSE.

**Compliance With Llm Reviewing Policy:**

Affirmed.

**Final Justification:**

Thank you to the authors for the detailed response. I agree that the theoretical analysis helps cover some experimental aspects. In light of this, I am happy to increase my score by one point to 3.

**Key Questions For Authors:**

How is the number of embedding dimensions decided? Will embedding dimension affect current results?

**Limitations:**

yes

**Strengths And Weaknesses:**

Strength 1: The writing is clear and mathematically grounded, which helps readers understand the background and problem setup. Additionally, theoretical guarantees for the proposed methods that satisfy stability are provided.

Strength 2: The proposed method is simple to implement and consistently produces strong results across 5 datasets and 9 baselines, as shown in Table 1.

Weakness 1: The performance of deep learning based approaches is known to be sensitive to hyperparameters. While the authors mention that the models are trained and evaluated using the standard, publicly documented protocols (Appendix L), it's important to note that the optimal hyperparameter setups could vary with datasets. When comparing to these baselines (JODIE, DyRep, TGN, and DyGFormer), hyperparameters (at least the key ones) should be optimized individually for each dataset to facilitate fair and meaningful comparison. Additionally, the details of hyperparameters used and the hyperparameter search range are required to be reported.

Weakness 2:
The t-SNE in Figure 1 is not very informative. While the authors claim that ULSE preserves temporal alignment better than TempCut-N, the visual differences are subtle, and the figure lacks annotations on which nodes share the same connectivity across two snapshots (for longitudinal) and which nodes share the same connectivity inside each snapshot (for cross-sectional). Please refer to Figure 1 of [1]. A similar point applies to Figure 4-7.

Weakness 3:
To demonstrate that ULSE satisfies both cross-sectional and longitudinal stability conditions, can the authors directly calculate some distance metrics? e.g., the paired displacement test defined in Sec. 4.2 in [1]

Weakness 4:
The performance of ULSE-n1 loses significantly to UASE when p<0.2 in Figure 3. Is this due to p<q? Additionally, current results show robustness only on p; a multivariate evaluation (on both p and q) would be more beneficial.

Weakness 5:
The experimental scope is narrow, as all downstream evaluation is based on node clustering. It would strengthen the paper to include additional tasks, such as node classification or link prediction, to assess whether the benefits of ULSE extend beyond clustering.

Weakness 6: Presentation concerns about formatting details and experimental setup:
6.1 The synthetic setup appears to contain three snapshots, but Figure 1 only visualizes two time steps.
6.2 Please remove (or be more specific) on the top captions in all the t-SNE figures. The meanings of the current titles, e.g., "USE-ULSE-n1 at Time 1" or "TemporalCut_fast_normalized_1_0 at Time 1", are unclear and duplicate the captions at the bottom.
6.3 Are the clustering metrics averaged over each snapshot?
6.4 Will the embedding dimensions of nodes affect results?

[1] Davis, E., Gallagher, I., Lawson, D. J., & Rubin-Delanchy, P. (2023). A simple and powerful framework for stable dynamic network embedding. arXiv preprint arXiv:2311.09251.

---

> ### Author Rebuttal · Authors · 2026-03-27
>
> We thank the reviewer for the careful reading and constructive feedback, and for recognizing the clarity and theoretical grounding of the paper. We address each point below.
>
> (1) Deep learning baselines / hyperparameters
>
> We agree that deep learning baselines can be sensitive to hyperparameters. We used the official implementations and publicly documented protocols (Appendix L) to ensure consistency and reproducibility, and in the revision we will explicitly clarify the key hyperparameters and selection protocol used in our experiments.
>
> More broadly, our goal was to compare against standard, reproducible evaluation protocols rather than introduce additional dataset-specific tuning variability across baselines. ULSE is an unsupervised embedding method designed to recover a latent structural representation with well-defined cross-sectional and longitudinal stability properties, whereas temporal deep baselines are typically trained with predictive objectives (e.g., link prediction) that do not explicitly target these properties. Accordingly, these baselines provide a useful practical comparison, but they are not specifically designed around the stability criteria that are central to our paper.
>
> (2) Figure 1 / t-SNE visualizations
>
> Thank you for this suggestion. We agree that the intended stability patterns could be communicated more clearly. In the synthetic visualizations, nodes belonging to the same community at the first snapshot are encoded using the same color and marker shape. The intended interpretation is that if a community’s connectivity pattern does not change across time, its embedded position should remain aligned (longitudinal stability), whereas if communities become connectivity equivalent at a later snapshot, their embeddings should coincide (cross sectional stability). Moreover, Figures 1 and 5 are not t-SNE, but direct plots of the first two embedding dimensions.
>
> We appreciate the pointer to Figure 1 of Davis et al. [1], and in the revision we will revise the synthetic visualizations to more directly indicate which nodes or communities are intended to illustrate longitudinal versus cross-sectional stability. We will also clarify the encoding in the captions and main text, simplify the titles, and improve the figure annotations so that the intended interpretation is more transparent.
>
> (3) Direct empirical stability metrics
>
> We agree that direct empirical diagnostics would strengthen the experiments. While the main contribution is theoretical, showing that ULSE satisfies cross sectional and longitudinal stability under the DSBM, we will discuss this perspective more explicitly in the revision, including paired displacement style analyses as a complementary diagnostic.
>
> (4) ULSE-n1 in the low p regime
>
> The degradation of ULSE-n1 for small $p$ is primarily due to extreme sparsity, rather than specifically $p < q$. Since ULSE-n1 uses per snapshot degree normalization, very sparse snapshots can lead to unstable normalization and poorer embeddings. ULSE-n2 is more robust in this regime because it uses partially aggregated normalization across time. We will clarify in the revision that the observed behavior is primarily driven by sparsity.
>
> (5) Scope of downstream tasks
>
> Additional downstream tasks such as node classification or link prediction would be interesting future extensions. Our current empirical focus is on clustering because the central theoretical claims of the paper concern recovery of a stable latent embedding geometry, specifically the cross sectional and longitudinal stability of node representations. Clustering is therefore the most direct downstream evaluation, since it tests whether nodes with equivalent structural roles are embedded consistently within and across time. We will clarify this motivation in the revision and note broader downstream evaluation as future work.
>
> (6) Presentation / setup details
>
> We will clarify the following in the revision:
>
> (6.1) the synthetic setup contains three snapshots, while Figure 1 shows two for clarity
>
> (6.2) figure titles will be simplified and standardized
>
> (6.3) clustering metrics are averaged across snapshots
>
> (6.4) The embedding dimension is tied theoretically to the rank of the underlying population signal (i.e., the number of informative singular directions), and in practice we use a standard spectral heuristic (e.g., an eigengap rule). This is distinct from the stability properties themselves, although the choice of dimension can influence the quality of the recovered embedding and downstream performance. In the revision, we will add a brief discussion of dimension sensitivity for completeness.
>
> Overall, we appreciate these suggestions and believe they will strengthen the paper. In the revision, we will improve empirical transparency and presentation while preserving the central contribution: a simple Laplacian based dynamic embedding framework with provable stability guarantees and strong robustness under degree heterogeneity.

---

> > ### Author Rebuttal · Reviewer_QMLw · 2026-04-02
> >
> > I thank the authors for the rebuttal. While the responses address the concerns point-by-point, they primarily acknowledge the issues and defer improvements to future revisions, without providing additional empirical evidence at this stage.
> >
> > Regarding hyperparameter sensitivity, while tuning is indeed standard practice for deep learning-based methods, it would strengthen the paper to include results using fixed hyperparameters (e.g., commonly adopted settings in prior work) in the main text, with more extensive tuning results presented in the Appendix.
> >
> > Although several presentation and experimental setup issues (Sections 3, 4, 6.1, 6.2, 6.3) appear relatively easy to address, other concerns (Points 1, 2, 3, 5, 6.4) seem to require more substantial revisions and additional experimental validation.

---

> > > ### Author Response · Authors · 2026-04-02
> > >
> > > Thank you again for the thoughtful follow-up and for distinguishing between issues that can be resolved through clarification and those that would require additional experiments beyond the rebuttal period.
> > >
> > > We agree that points (2), (4), (6.1), (6.2), and (6.3) are primarily matters of presentation and experimental specification, and we will address these directly by clarifying the synthetic setup, improving the figure annotations/titles, and stating explicitly that clustering metrics are averaged across snapshots.
> > >
> > > For the remaining concerns, we would like to respectfully clarify the distinction between core validation and broader empirical scope. The central contribution of the paper is a normalized-Laplacian extension of unfolded spectral embedding with provable cross-sectional and longitudinal stability, together with consistency guarantees and a Laplacian-specific spectral interpretation. Our current experiments are designed to evaluate this contribution in the downstream setting most directly tied to the theory, namely clustering of structurally equivalent nodes across time.
> > >
> > > In that context, we view point (5) on additional downstream tasks as a valuable extension of scope rather than a prerequisite for supporting the paper’s main claims. Similarly, point (6.4) concerns practical model selection and completeness, rather than the validity of the stability results themselves.
> > >
> > > For point (1), our intention was to compare against standard, publicly documented protocols in a reproducible and consistent way, rather than introduce dataset-specific tuning choices across methods with different training objectives. We agree that making the key hyperparameter choices explicit would improve transparency, and we will do so.
> > >
> > > For point (3), we agree that direct empirical stability diagnostics would provide a useful complementary view. At the same time, the primary evidence for stability in the paper is theoretical: the stability properties are established under the DSBM, while the empirical section is designed to show that the resulting embeddings are useful in the downstream task most directly aligned with those guarantees.
> > >
> > > We are grateful for these comments and will use them to strengthen both the presentation and the empirical framing of the paper.

---

### Decision · Program_Chairs · 2026-04-30

**Decision:**

Accept (regular)

**Comment:**

This paper presents a method for node embeddings in dynamic graphs based on the Unfolded Laplacian. While the reviewers appreciated the overall idea and theoretical contribution with the specialized Cheeger's inequality, they raised quite significant concerns about the novelty of the method amidst recent works and the overall "unrolled" literature, as well as limited experimental settings and complexity/scalability of the method.